



**Impacts of Thermodynamic and Dynamic Processes on the Vertical**
**Distribution of Carbonaceous Aerosols: lessons from in-situ**
**observations at eastern foothills of LiuPan Mountains, Loess Plateau**
Shaofeng Qi[1,2], Suping Zhao [1,3*], Ye Yu[1,3], Longxiang Dong[1,3], Tong
Zhang[1,3], Guo Zhao[1,2,3], Jianglin Li[1,3], Xiang Zhang[1,2], Yiting Lv[1,2]
1. Key Laboratory of Cryospheric Science and Frozen Soil Engineering, Northwest
Institute of Eco-Environment and Resources, Chinese Academy of Sciences, Lanzhou
730000, China.
2. University of Chinese Academy of Sciences, Beijing 100049, China.
3. Pingliang Land Surface Process & Severe Weather Research Station, Pingliang
744015, China.
*Correspondence to*: Suping Zhao (zhaosp@lzb.ac.cn)
**Abstract.** The vertical distribution of carbonaceous aerosols critically influences
planetary boundary layer structure and climate impacts. However, high-resolution
vertical data remain scarce over the Chinese Loess Plateau. To address this gap,
coordinated observations of carbonaceous aerosols and meteorological variables were
conducted in the Loess Plateau using tethered balloon-borne instruments during two
field campaigns in July 2023 and 2024. The average near-surface concentrations of
black carbon (BC) and ultraviolet particulate matter (UVPM) in Pingliang were 0.82
µg m⁻³ and 1.26 µg m⁻³, respectively. Vertically, carbonaceous aerosol concentrations
generally decreased with height. A comparison of the vertical profiles of BC, UVPM,
VTKE (mechanical turbulence), and potential temperature showed that during the
early morning and nighttime, when convective activity was weak, UVPM
concentrations in the upper atmosphere were higher than those of BC. This pattern is
primarily attributed to nucleation processes involving gaseous precursors during
nighttime. Analysis of the roles of dynamic and thermodynamic processes indicated





that thermodynamic processes dominated aerosol vertical transport in the near-surface
layer, while enhanced dynamic processes at higher altitudes facilitated horizontal
dispersion of pollutants. Air masses from the south of the observation site contributed
significantly to UVPM levels. As air mass altitude decreased, the influence of local
sources became more pronounced. Overall, this study demonstrated the regulatory
mechanism of daytime and nighttime thermodynamic and dynamic impacts on the
vertical distribution of pollutants.

## 1. Introduction

Carbonaceous aerosols, particulate matter generated from fossil fuel combustion and
biomass burning, directly impact the Earth-atmosphere energy budget by absorbing
solar radiation. Their primary components are organic carbon (OC) and black carbon
(BC). OC encompasses both primary organic carbon (POC) emitted directly from
biomass burning and secondary organic carbon (SOC) formed through the
photochemical reactions of volatile organic compounds (VOCs). Due to its complex
chemical composition, OC introduces substantial uncertainty in climate effect
assessments (Kroll et al., 2011). Black carbon (BC), characterized by strong solar
radiation absorption, represents the second-largest anthropogenic climate forcer after
$CO_2$ (Bond et al., 2013; Ramanathan & Carmichael, 2008). Near the base of the
stratosphere, BC's direct radiative forcing can be approximately ten times stronger
than at the surface (Samset & Myhre, 2011). Consequently, its vertical position within
the atmosphere significantly influences atmospheric stratification (Zhang et al., 2017;
Zhao et al., 2021). Within the surface layer, BC can heat the lower atmosphere,
enhancing convection and promoting boundary layer development. At higher
altitudes, however, BC heats the atmosphere while simultaneously reducing solar
radiation reaching the surface, leading to increased atmospheric stability. This
suppresses boundary layer development and exacerbates air pollution (Ding et al.,
2016; Petäjä et al., 2016; Wang et al., 2018a). Over complex terrain, valley wind
systems and orographically-induced turbulence can transport surface-emitted BC to



higher elevations, where it accumulates within temperature inversion layers.
Absorptive BC further heats the atmosphere, suppressing the development of the
planetary boundary layer (PBL) and altering its height and stability (Zhao et al.,
2023). Additionally, BC modifies cloud microphysical properties, influencing cloud
formation, dissipation, and precipitation, thereby affecting regional and global climate
systems (Panicker et al., 2014; Wendisch et al., 2008). Over recent decades, the
radiative forcing and climate effects of BC have been extensively studied. Published
estimates of BC's direct radiative forcing range between 0.25–0.90 W m$^{-2}$ (Allen &
Landuyt, 2014), yet these values are subject to considerable uncertainty. A primary
source of this uncertainty is the significant spatiotemporal heterogeneity of BC at the
global scale, stemming from regional combustion processes and its short atmospheric
lifetime. Consequently, using default model BC profiles to simulate radiative forcing
introduces substantial errors (Hodnebrog et al., 2014). Reported studies indicate that
differences in BC vertical distribution contribute approximately 20–40% to the
uncertainty in calculations of BC's top-of-atmosphere radiative forcing (Chen et al.,
2022; Zarzycki & Bond, 2010). Therefore, accurately characterizing the true vertical
distribution of BC in the atmosphere is crucial for the precise assessment of its
regional and global climate effects.

However, acquiring accurate BC vertical profiles remains a significant challenge for
researchers globally. Current observation methods include: Tethered balloons (Guan et
al., 2022; Wang et al., 2021a; Zhao et al., 2023), Aircraft measurements (Moorthy et
al., 2004; Schwarz et al., 2017), Unmanned Aerial Vehicles (UAVs) (Liu et al., 2020;
Wang et al., 2021b; Wu et al., 2021), Cable cars (Zawadzka et al., 2017),
Meteorological towers (Wang et al., 2018b; Xie et al., 2019), Topography-dependent
in-situ observations (Zhao et al., 2019; 2022). Notably, aircraft measurements face
operational constraints in complex terrain due to high costs and requirements for
expansive landing areas. UAVs, cable cars, and meteorological towers are limited by
maximum achievable altitudes – most UAVs typically reach only ~500 m, while cable
cars and towers cover even lower vertical ranges. Though topography-dependent



87 observations are widely applied in complex terrain, their coarse spatial resolution

88 cannot resolve true vertical distributions of atmospheric constituents. Additionally,

89 while lidar remote sensing enables continuous profiling (Miffre et al., 2015), its

90 accuracy remains inferior to in-situ techniques. In contrast, tethered-balloon systems

91 mitigate these key constraints by providing high-resolution BC profiles within the

92 planetary boundary layer. This approach has been successfully deployed across

93 diverse regions including Europe, the Arctic, South Asia, and China's North China

94 Plain, Sichuan Basin, and Yangtze River Delta (Bisht et al., 2016; Mazzola et al.,

95 2016; Ran et al., 2016; Samad et al., 2020; Wang et al., 2018b).

96

97 Pingliang City, situated in eastern Gansu Province, lies at the convergence of Shaanxi,

98 Gansu, and Ningxia provinces. As a significant component of the Loess Plateau and a

99 major agricultural zone in Northwest China, understanding its climatic-environmental

100 mechanisms is crucial for improving meteorological forecasting accuracy and

101 formulating effective pollution control strategies. The region's complex topography

102 facilitates the transport of urban pollutants to higher-elevation loess tablelands via

103 valley wind circulations, enabling dispersion during transit. Furthermore, influenced

104 by the towering Liupan Mountains, Pingliang exhibits pronounced vertical climatic

105 heterogeneity, resulting in intricate feedback mechanisms between the planetary

106 boundary layer and aerosols. Current research reveals scarce data on the vertical

107 distribution of BC from fossil fuel combustion, biomass burning, and mineral dust

108 activities in this region. To address this gap, we conducted detailed vertical profile

109 observations using tethered balloon-borne instrumentation over typical loess tableland

110 areas during July 2023 and July 2024. This study aims to supplement the deficiency in

111 local BC aerosol observations, establish a database for assessing aerosol climate

112 effects on the Loess Plateau, and ultimately provide scientific foundations for

113 pollution control strategies.

114

115 **2. Materials and methods**



**2.1 Observation methods and data sources**

Field observations were conducted at the Pingliang Land Surface Processes and
Severe Weather Research Station (Pingliang Station), Chinese Academy of Sciences,
during two intensive campaigns from 15 to 24 July 2023 and 17 to 30 July 2024. The
station is located in Baimiao Township, Pingliang City, Gansu, with geographic
coordinates detailed in Figure 1. Near-surface meteorological parameters at Pingliang
Station and the Air Quality Index (AQI) for Pingliang City during the observation
periods are presented in Figure S1, while Figure S2 displays concentrations of $PM_{10}$,
$PM_{2.5}$, CO, $SO_2$, $NO_2$, and $O_3$ in Pingliang's urban area. Meteorological data were
obtained from near-surface observations at Pingliang Station, and air quality
measurements originated from the Pingliang Environmental Monitoring Station
(Station ID: 2656A; http://eia-data.com/). Results indicate that southeasterly or
northwesterly winds prevailed at Pingliang Station during the observation periods,
with wind speeds typically below 1 m s$^{-1}$. The mean temperature was 20.76°C and
mean relative humidity was 76.84%. Air quality in Pingliang remained generally good
except during sporadic dust pollution events.

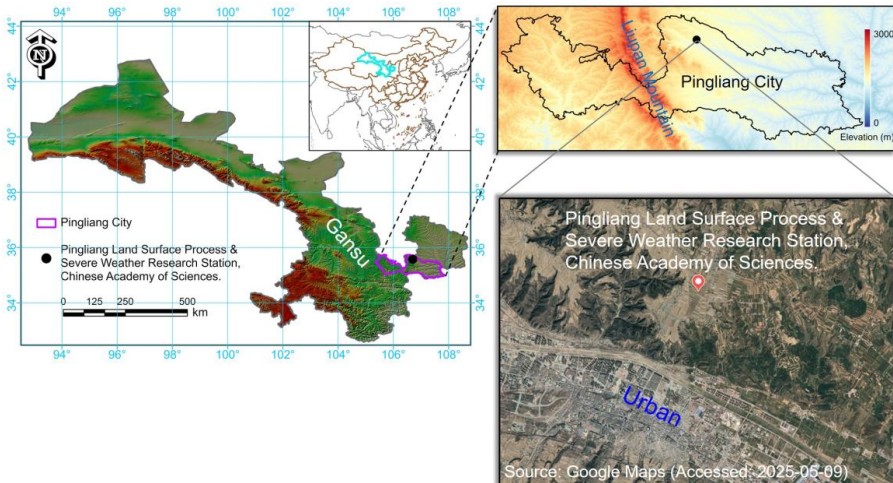

Figure 1. Geographic location of the observation site. The map is a pure reproduction of Google
Maps with added marks for our study locations. Copyright © Google Maps. Publisher's note:
Please note that the above figure contains disputed territories.



2.1.1 Tethered balloon platform
Vertical profiling was primarily conducted using instrumentation suspended from a
tethered balloon system. The system comprised a 10 m³ helium-filled balloon, a tether
line, and an electrically powered winch that actively regulated both ascent and descent
rates. Buoyancy provided initial lift, while the winch precisely controlled vertical
maneuvering. Instruments were mounted 30 m below the balloon to minimize direct
atmospheric disturbance. A MicroAeth® MA350 aerosol monitor measured vertical
distributions of carbonaceous aerosols, and an iMet-4 radiosonde (iMet, USA)
acquired temperature and humidity profiles. Observations were conducted at 3-hour
intervals from 05:00 to 23:00 local time (05:00, 08:00, 11:00, 14:00, 17:00, 20:00,
23:00) daily. Operations were suspended during adverse conditions (e.g., strong winds
or precipitation). Complementary vertical wind profiles were obtained using Doppler
Beam Steering (DBS) mode of a Leosphere Windcube 200s lidar. Each balloon
sounding lasted approximately 30–60 minutes, during which atmospheric conditions
remained relatively stable, allowing both ascent and descent paths to characterize
vertical structures. A total of 24 and 39 soundings were completed during summer
2023 and 2024 respectively, yielding 126 vertical atmospheric profiles.

2.1.2 Carbonaceous aerosol mass concentrations
The MicroAeth® MA350 determines BC concentration based on the Beer-Lambert
law, quantifying light absorption by carbonaceous particles deposited on a PTFE filter
at multiple wavelengths. The instrument employs five laser wavelengths: 375 nm
(UV), 470 nm (Blue), 528 nm (Green), 625 nm (Red), and 880 nm (IR). Carbon
concentration measured at 880 nm is considered equivalent to BC concentration
(denoted as IRBC or BC) (Zhao et al., 2023), while measurements at 375 nm
represent Ultraviolet Particulate Matter (UVPM) associated with biomass combustion
(e.g., wood, straw). The Ångström Absorption Exponent (AAE), calculated from
multi-wavelength measurements, characterizes the wavelength dependence of
carbonaceous aerosol absorption. During this study, the MA350 operated at a flow
rate of 150 mL min⁻¹ with a 5-second sampling interval. Negative optical attenuation



(ATN) values occasionally occurred under low aerosol concentrations at high
altitudes, which were corrected using the Optimized Noise-reduction Averaging
(ONA) algorithm (Hagler et al., 2011).

**2.2 Calculation of light absorption coefficient for carbonaceous aerosols**
The light absorption coefficient ($b_{Abs}$) of black carbon was calculated using the
following Eq. (1):
$b_{Abs} = MAC_{BC}(\lambda) \times [BC]$ (1)
where $MAC_{BC}(\lambda)$ denotes the mass absorption cross-section of BC at specific
wavelengths. For the MicroAeth® series instruments, the MAC values at 375 nm, 470
nm, 528 nm, 625 nm, and 880 nm are 24.07 m² g⁻¹, 19.07 m² g⁻¹, 17.03 m² g⁻¹, 14.09
m² g⁻¹, and 10.12 m² g⁻¹, respectively (Zhao et al., 2023). The [BC] represents the
mass concentration of BC.

Additionally, the light absorption coefficient of BC can be expressed as:
$b_{Abs}(\lambda) = k \times \lambda^{-AAE}$ (2).
In Eq. (2), k is a wavelength-independent constant, and AAE represents the Ångström
Absorption Exponent, which characterizes the wavelength dependence of BC's light
absorption. A higher AAE value signifies that the aerosol absorption capacity
decreases more rapidly with increasing wavelength. By combining the $b_{Abs}$ values
derived from Eq. (1) with Eq. (2), vertical profiles of AAE for BC were obtained.

**2.3 Determination of planetary boundary layer height**
Potential temperature ($\theta$, in K), defined as the temperature an air parcel would attain if
adiabatically brought to a standard pressure level (Han et al., 2019; Seidel et al.,
2010), is calculated as Eq. (3):
$\theta = (T+273.15) \times (\frac{P_0}{P})^{\frac{R_d}{C_{pd}}}$ (3)
where T denotes measured air temperature (°C), P represents air pressure (hPa), $P_0$ is
the standard reference pressure (1000 hPa), $R_d$ signifies the gas constant for dry air





(287 J kg⁻¹ K⁻¹), and $C_{pd}$ corresponds to the specific heat of dry air at constant
pressure (1005 J kg⁻¹ K⁻¹).

Specific humidity (q, g g⁻¹), defined as the ratio of water vapor mass to the total mass
of moist air (water vapor plus dry air), is calculated using Eq. (4),
$$q = \frac{\varepsilon \times e}{P - 0.378 \times e} \tag{4}$$
where $\varepsilon = 0.622$, e represents vapor pressure, and P denotes atmospheric pressure,
with vapor pressure, e being derived from Eq. (5) through relative humidity (RH) and
air temperature (T, °C).
$$e = 6.105 \times RH \times \exp\left(\frac{17.7 \times T}{237.7 + T}\right) \tag{5}.$$

The planetary boundary layer (PBL) refers to the lower troposphere directly
influenced by surface forcing with a response time of less than one hour, playing a
critical role in the dispersion and transport of air pollutants. Within the PBL, turbulent
mixing processes homogenize air temperature and humidity, resulting in relatively
uniform distributions of these properties. Distinct discontinuities in temperature,
humidity, and wind speed typically mark the PBL top (Emeis et al., 2008; Seibert et
al., 2000). While various methodologies exist for determining PBL height (PBLH),
this study employs two established approaches: the potential temperature gradient
method and the parcel method, with detailed computational procedures provided in
Supplementary Table S1 (Zhang et al., 2020)

**2.4 Impacts of thermodynamic and dynamic processes on vertical distribution of**
**carbonaceous aerosols**
To quantify the relative contributions of potential temperature gradient, mechanical
turbulence index, horizontal wind speed, and vertical wind speed to UVPM variations
at different altitudes, we employed a random forest regression algorithm. The model
generated training subsets via bootstrap sampling, with random feature subsets
selected for optimal splitting at each decision tree node. Observations were



categorized into daytime (08:00, 11:00, 14:00, 17:00 LT) and nighttime (20:00, 23:00,
05:00 LT) periods to compare the dominant mechanisms governing aerosol vertical
distribution. The mechanical turbulence index was calculated using Eq. (6).
$\mathrm{V_{TKE}} = 0.5 \times \sqrt{\overline{u^2} + \overline{v^2} + \overline{w^2}}$ (6).

**2.5 Identification of potential source regions**
In this study, GDAS1 meteorological data (1° × 1° resolution) obtained from the
NOAA FTP repository (ftp://arlftp.arlhq.noaa.gov/pub/archives/gdas1) were utilized.
These data were processed using the MeteoInfo software to calculate backward
trajectories and perform cluster analysis (http://www.meteothink.org/). Subsequently,
the potential source contribution function (PSCF) and concentration weighted
trajectory (CWT) analysis toolkits within MeteoInfo were employed to identify
potential source regions and quantify their relative contributions (Wang, 2014).

**3. Results and discussions**
**3.1 Vertical profiles of carbonaceous aerosols**
3.1.1 General characteristics of BC
Observations indicate that near-surface mass concentrations of BC and UVPM
averaged 0.82 μg m⁻³ and 1.26 μg m⁻³, respectively. During the campaign, UVPM
concentrations varied between 0.21 and 5.63 μg m⁻³ across different altitudes,
whereas BC ranged from 0.13 to 2.05 μg m⁻³. Compared with previous studies
conducted in other regions of China, the concentration of BC observed in Pingliang is
lower than those reported in Beijing, Shanghai, Nanjing, Chengdu, Shenzhen,
Hengshui, the Beibu Gulf region, and Lanzhou (Guan et al., 2022; Ran et al., 2016;
Shi et al., 2021; Wang et al., 2021a; Wu et al., 2021; Yang et al., 2023; Yang et al.,
2022; Zhao et al., 2023). This discrepancy can be attributed to several factors. Firstly,
Pingliang is a relatively small city with a permanent population of fewer than 2
million, whereas the aforementioned cities have much larger urban populations.
Consequently, the total amount of air pollutants generated from daily human activities



in Pingliang is comparatively lower. Secondly, the observation site in Pingliang is
situated at a higher elevation than the urban center, thereby reducing the influence of
direct urban emissions. In contrast, observation sites in other cities are typically
located in suburban areas that are more directly affected by emissions from urban
cores. Wang et al. (2019) conducted tethered-balloon measurements of the vertical
distribution of BC over the Tibetan Plateau and found even lower BC concentrations
than those in the present study. Furthermore, the rate of decrease in BC concentrations
with altitude was more pronounced at the Plateau site compared to Pingliang. Overall,
existing studies consistently indicate that BC concentrations decrease with increasing
altitude. However, due to differences in terrain, PBLH, and atmospheric diffusion
conditions, the vertical profiles of BC vary significantly among different sites. For
instance, in Shanghai, Wang et al. (2021a) reported a sharp decline in BC
concentrations at around 600 m in the morning, while in the afternoon, the decrease
occurred at approximately 800 m due to stronger convective mixing. Over the Tibetan
Plateau, BC concentrations dropped markedly at altitudes as low as 100–200 m (Wang
et al., 2019). Yang et al. (2023) and Zhao et al. (2023) reported elevated BC
concentrations near 2000–2500 m, which they attributed to the combined effects of
upper-level subsidence and lower-level updrafts. Similarly, Lu et al. (2019) and Chen
et al. (2022) observed elevated BC concentrations in the 500–800 m layer over Anhui
and Beijing, respectively, primarily influenced by the presence of upper-level
temperature inversions.

From the global perspectives, the near-surface BC concentration in Delhi, India, was
reported to be approximately 30.00 μg m⁻³, which is substantially higher than those
observed in China and Europe (Bisht et al., 2016). In contrast, near-surface BC
concentrations in Stuttgart, Germany, and Milan, Italy, were found to be comparable
to those in Shenzhen, China, at around 2.00–3.00 μg m⁻³. These values are lower than
those reported in other Chinese cities such as Shanghai, Nanjing, Chengdu, Hengshui,
and Lanzhou, but still higher than the concentrations observed in this study (Ferrero et
al., 2011; Samad et al., 2020). In the Arctic region, due to limited anthropogenic



influence, near-surface BC concentrations are generally lower than those observed
both over the Tibetan Plateau in China and in the Loess Plateau region investigated in
this study (Ferrero et al., 2016).

3.1.2 Diurnal variations of BC and UVPM profiles
Figures S3 and S4 respectively depict the trends and correlations of the ascent and
descent profiles for BC and UVPM. The results indicate that ascent and descent
profiles in this study exhibit very similar trends with only minor differences,
warranting their combination into a single mean profile. Figure 2 shows the averaged
profiles for all sampling periods during the observation campaign, including both
ascent and descent legs of the tethered balloon. In Figure 2, the red solid line and its
shaded envelope denote the mean and standard error of IRBC, while the blue solid
line and its shaded envelope denote the mean and standard error of UVPM.
Furthermore, owing to the relatively high wind speeds in the upper atmosphere,
daytime measurements in this study generally did not reach the top of the PBL.

Vertical profiles of BC and UVPM concentrations reveal a consistent decrease with
increasing altitude, with the most pronounced gradient observed in the near-surface
layer. A comparative analysis of their vertical profiles reveals that during periods of
weak convective activity, such as early morning and nighttime, UVPM concentrations
aloft generally exceed those of BC, while near the surface the two species show much
smaller differences. This elevated UVPM-to-BC ratio at altitude may be attributable
to the lighter molecular weight and smaller size of gas-phase precursors compared to
soot particles. Under nocturnal stable stratification, vertical mixing is suppressed,
allowing volatile organic compounds (VOCs) to be lofted more readily into the free
troposphere. There, in a relatively clean atmosphere lacking efficient coagulation
sinks, $NO_3^-$ dominated nighttime chemistry and low temperatures enhance secondary
organic aerosol formation via low-temperature condensation and gas–particle
partitioning (Han & Jang, 2023; Kuang et al., 2025; Kulmala, 2022; Morgan et al.,
2009; Wang et al., 2023; Zhao et al., 2024). These processes collectively amplify the



UVPM–BC concentration difference at night. After sunrise, increased solar heating
invigorates convection and erodes the nocturnal inversion at the boundary-layer top.
Pollutants trapped in the residual layer are then entrained and mixed downward into
the daytime boundary layer, reducing the UVPM–BC disparity while both profiles
continue to decline with altitude (Zhao et al., 2023). Notably, between 14:00 and
17:00 LST, strong convective mixing homogenizes BC and UVPM within the 100–
800 m layer; however, BC is not effectively transported to higher altitudes, resulting
in a pronounced UVPM–BC difference aloft by 17:00. By approximately 20:00 LST,
as convective activity wanes, the UVPM–BC concentration difference reemerges
throughout the entire column and remains more pronounced at higher elevations than
near the surface.

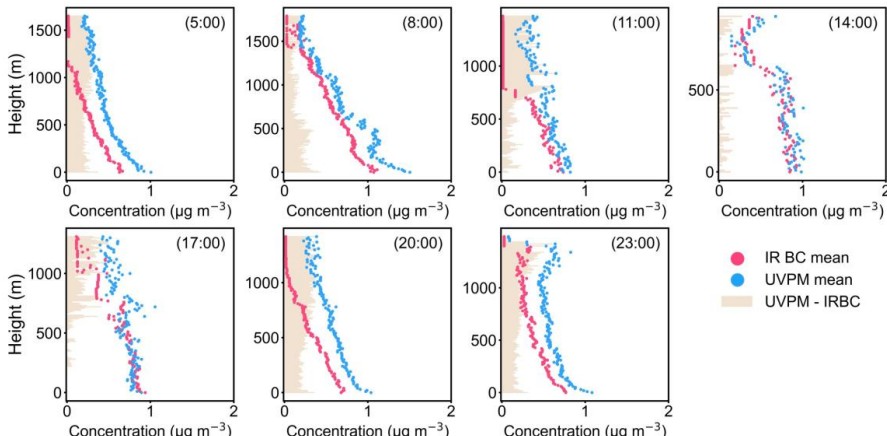


Figure 2. Diurnal variation of the average IRBC and UVPM concentration profiles in the
Pingliang region. The red and blue curve represents IRBC and UVPM concentrations,
respectively. The light beige shaded area represents the difference between UVPM and IRBC.

To better characterize BC vertical structure within the stable boundary layer, the
observed profiles were classified into four types, each exhibiting distinct features. In
Cluster 1, BC and UVPM decrease almost uniformly with altitude at a rate of
approximately 0.51 μg m⁻³ km⁻¹ up to 1000 m. Cluster 2 also shows comparable near-
surface concentrations of both species, but with a rapid decline of about 1.23 μg m⁻³
km$^{-1}$ below 250 m followed by a more gradual decrease above, reflecting a stronger
nocturnal inversion that inhibits upward diffusion. Cluster 3 profiles, typically
observed at 05:00, 08:00, and 20:00 LST under intense inversions, display a modest
decrease of approximately 0.10 μg m$^{-3}$ km$^{-1}$ from the surface to 100 m, an unexpected
increase in BC and UVPM between 100 and 600 m (attributable to pollutant
accumulation and upward mixing within the deep neutral residual layer; Kulmala et
al., 2023), and a decline above 600 m. In Cluster 4, concentrations remain nearly
constant below 200 m but rise with height above this level, likely due to a weak
neutral stratification between 200 and 400 m. Because these observations coincided
with strong upper-level winds, measurements above approximately 400 m were
curtailed to protect instrumentation, leaving open the question of whether Cluster 4
would exhibit a high-altitude decrease similar to Cluster 3.

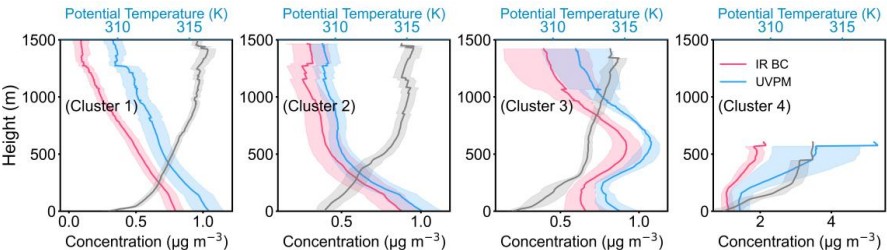


Figure 3. Cluster analysis of BC concentration profiles in the Pingliang region. The red curve
represents IRBC, while the blue curve represents UVPM. The red and blue shaded areas
indicate the standard errors of IRBC and UVPM, respectively.

3.1.3 Diurnal variations of AAE profiles
Figure S5 presents the variations of the light absorption coefficients of UVPM and BC
at different altitudes. Overall, both UVPM and BC absorption coefficients decline
steadily with increasing altitude, a pattern primarily controlled by the mass
concentrations of carbonaceous aerosols. Moreover, the difference between the
UVPM and BC absorption coefficients diminishes with height, indicating that UVPM
dominates light absorption by carbonaceous aerosols in the lower atmosphere,
whereas BC exerts a greater influence aloft, consistent with the findings of Qi et al.



(2025). The AAE, which characterizes the wavelength dependence of the mass-
specific absorption by BC (calculation details are given in Figure S6), was calculated
for the study period; the average diurnal AAE values are shown in Figure S7.
Compared with Wu et al. (2021), the AAE of BC in Pingliang (1.65) is substantially
higher than that reported for Shenzhen (<1.00). We further selected observations from
representative pollution events to compare AAE under different weather conditions
(Figure 4). During dust episodes, AAE increases markedly, by approximately 83%
relative to dust-free conditions. Likewise, emissions from diesel vehicles yield higher
AAE than periods without diesel contributions. Under heavy fog in the lowest 200 m,
AAE is also elevated; at 200 m the rapid decrease in AAE likely results from the
sharp reduction of water vapor aloft. Hence, compared with daytime, the lower
frequency and intensity of anthropogenic activity at nighttime lead to a narrower
range of aerosol sources and a smaller vertical variation in AAE across the entire
profile.

Previous studies commonly employ a two-component model to differentiate the
absorption Ångström exponent of black carbon ($AAE_{BC}$) and brown carbon ($AAE_{BrC}$),
fixing $AAE_{BC}$ at 1 and thereby deriving $AAE_{BrC}$ (Figure S8). An $AAE_{BrC}$ value greater
than 2.0 is generally attributed to biomass burning and secondary aging processes.
The diurnal $AAE_{BrC}$ profiles shown in Figure S8 reveal pronounced contributions
from secondary organic aerosol formation or from mixed emissions of biomass and
fossil fuels at our observation site. Overall, daytime $AAE_{BrC}$ values exceed those
recorded at night and in the early morning, reflecting the stronger photochemical
activity during daylight hours that promotes carbonaceous aerosol aging.



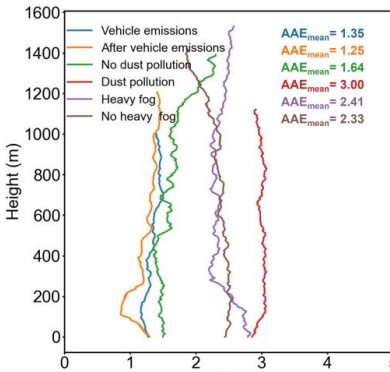

Figure 4. AAE profiles under different pollution conditions. (Vehicle emissions, After vehicle emissions, No dust pollution, Dust pollution, Heavy fog and No heavy fog occurred at 20:00 on July 17, 2024; 23:00 on July 17, 2024; 08:00 on July 20, 2024; 08:00 on July 21, 2024; 05:00 on July 25, 2024; and 05:00 on July 26, 2024, respectively.)

**3.2 Thermodynamic impacts on profiles of BC and UVPM**

Diurnal evolution of the PBLH is one of the primary factors controlling aerosol vertical distribution and is essential for understanding feedbacks between the boundary layer meteorology and aerosols. Zhang et al. (2020) reviewed and compared multiple PBLH retrieval methods. From the perspective of tracer distributions, PBLH inferred from vertical profiles of tracers such as water vapor and aerosols is termed the material PBLH ($PBLH_C$; Shi et al., 2020). Alternatively, PBLH can be derived from the vertical gradient of potential temperature ($PBLH_\theta$), with calculation protocols summarized in Table S1. Because PBLH evolves continuously, we selected a day with uninterrupted observations; however, owing to strong upper-level winds, continuous carbonaceous aerosol profiles were obtained only on 27 July 2024 at seven time slots (Figure 5). Hence, this day serves as a case study for examining how diurnal PBL evolution influences aerosol vertical structure. The parcel method is well suited to convective conditions but cannot accurately resolve $PBLH_\theta$ during early morning and nighttime (Holzworth, 1964), whereas it performs reliably under strong daytime convection. Using this approach, $PBLH_\theta$ at 11:00 and 17:00 LST were 508 m and 450 m, respectively; at 14:00, measurement ceilings did not reach $PBLH_\theta$. By



contrast, the potential-temperature gradient method yields accurate estimates at night
and in the early morning: $PBLH_\theta$ at 05:00, 08:00, 20:00, and 23:00 LST were 260 m,
181 m, 260 m, and 223 m, respectively. A comparison of UVPM and BC profiles with
these $PBLH_\theta$ values reveals that nighttime $PBLH_\theta$ from potential-temperature
gradients exceeds the $PBLH_C$ inferred from aerosol distributions, a phenomenon also
noted by Jiang et al. (2021). This discrepancy arises because nocturnal longwave
radiative cooling and weakened turbulence enhance near-surface stability and often
produce inversions; during their early development, $PBLH_\theta$ determined by potential-
temperature gradients tends to overshoot the inversion top, whereas $PBLH_C$ more
closely aligns with the inversion height.

Analysis of the potential-temperature profiles in Figure 5 indicates that at around
05:00 LST the planetary boundary layer top lay near 260 m. Below this altitude, both
BC and UVPM concentrations decrease slightly with height. The potential-
temperature profile further reveals a deep residual layer above the boundary-layer top,
where colder near-surface air is trapped beneath warmer air aloft, creating a stable
stratification that inhibits mixing within that layer and leads to increasing particle
concentrations toward its base. In the transition to the free troposphere above the
residual layer, comparatively low aerosol concentrations and enhanced turbulence
promote further dilution, and beyond approximately 500 m BC and UVPM again
decline with height. By 08:00 LST the boundary-layer height remained near 200 m,
and the vertical variation in aerosol concentrations mirrored the pattern observed at
05:00. With increasing solar insolation, however, surface heating intensified
convection so that by 11:00 LST the boundary-layer top had risen to roughly 500 m.
During this stage, relatively small vertical gradients in BC and UVPM within the
boundary layer indicate well-mixed conditions. At 14:00 LST tethered-balloon
sampling did not reach the boundary-layer top, but observations within the layer show
uniform aerosol distributions, preventing a direct assessment of boundary-layer-height
effects on concentration profiles. The potential-temperature gradients between 11:00
and 14:00 LST exhibit significant fluctuations, signaling unstable stratification



favorable to vertical pollutant transport (Li, 2019). From 14:00 to 17:00 LST, as solar
radiation waned and surface temperatures fell, the boundary-layer top subsided to
about 450 m. At this time, potential temperatures within the boundary layer remained
lower than at the surface and displayed pronounced variability, reflecting continued
unstable stratification and strong vertical mixing; BC and UVPM maintained nearly
uniform distributions. Above the boundary layer, a mixed layer approximately 300 m
thick persisted; at its top, diminished turbulence inhibited aerosol dispersion, causing
localized accumulation of BC and UVPM (Ding et al., 2016). By 20:00 LST, sunset-
driven surface cooling weakened convection, a nocturnal inversion developed near the
ground, and calm winds led to pollutant accumulation at low altitudes. As surface
temperatures continued to drop, vertical transport further diminished, confining
aerosols below roughly 200 m by 23:00 LST.

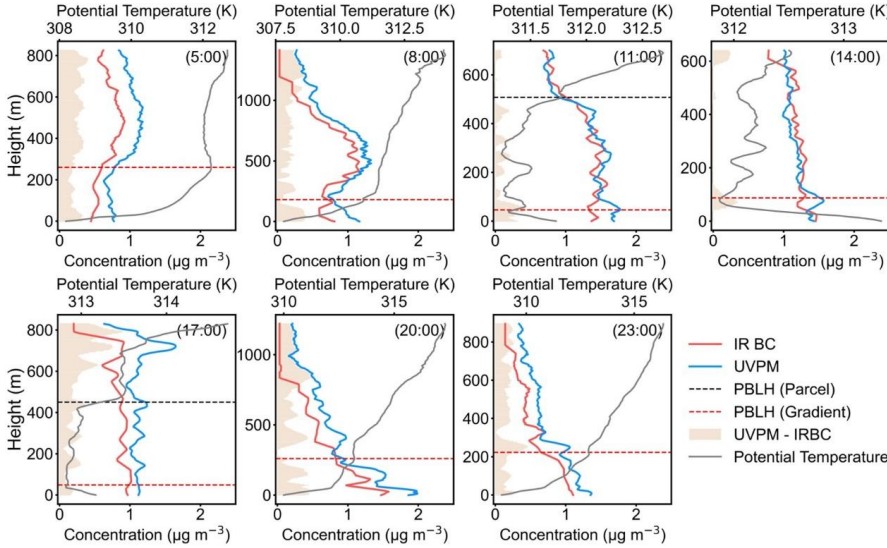


Figure 5. Diurnal variations of BC, UVPM and potential temperature profiles on July 27,
2024. The red curve represents IRBC, the blue curve represents UVPM, and the gray curve
represents the potential temperature profile. The light beige shaded area represents the
difference between IRBC and UVPM. The black and red dashed lines represent the PBLH
calculated using the air parcel method and the potential temperature gradient method,
respectively.




**3.3 Dynamic impacts on profiles of BC and UVPM**

3.3.1 Impacts of long-range transport on carbonaceous aerosols

Long-range transport of air masses plays a crucial role in shaping the vertical

distribution of air pollutants. Figure S9 presents the 500 m trajectories and their

altitude profiles for air masses arriving at the site during the observation period. It

shows that, upon entering the Pingliang region, these air parcels generally descend to

below 1500 m, meaning that, in addition to pollutants carried within the air mass

itself, emissions from surrounding urban areas also significantly impact the receptor

site. Trajectory-cluster analysis at 100 m, 500 m, and 1000 m (Figure 6) reveals that,

in summer, Pingliang is principally influenced by air masses originating from Inner

Mongolia and Ningxia to the north, from Gansu–Qingyang and northern Shaanxi to

the east, and from southern Shaanxi to the south. At 500 m and 1000 m, regional

contributions from these directions are broadly similar, whereas at 100 m the

proportion of short-range flow from the southeast increases markedly. The shaded

overlays in Figure 6 show the potential source contribution function (PSCF) and

concentration-weighted trajectory (CWT) results for UVPM. PSCF indicates that air

parcels from Inner Mongolia, Ningxia, Shanxi, and Shaanxi to the north, east, and

south exert the greatest influence on the Pingliang site. Specifically, at 100 m the

highest PSCF values occur along the Shaanxi–Gansu border, while at 500 m parcels

from central Shaanxi and the Shanxi–Henan–Shaanxi nexus dominate, followed by

the tri-provincial junction of Shaanxi, Gansu, and Sichuan. However, the CWT

analysis identifies Hanzhong in southern Shaanxi as the most significant source region

for UVPM at the receptor. Taken together, PSCF and CWT pinpoint southern Shaanxi

cities and local emissions around Pingliang as major contributors to carbonaceous

aerosol pollution, whereas at higher altitudes UVPM is primarily transported from the

south. Overall, these source-apportionment results align with the conditional

probability function (CPF) analysis, confirming that southerly flows contribute most

substantially to UVPM concentrations at the observation site.





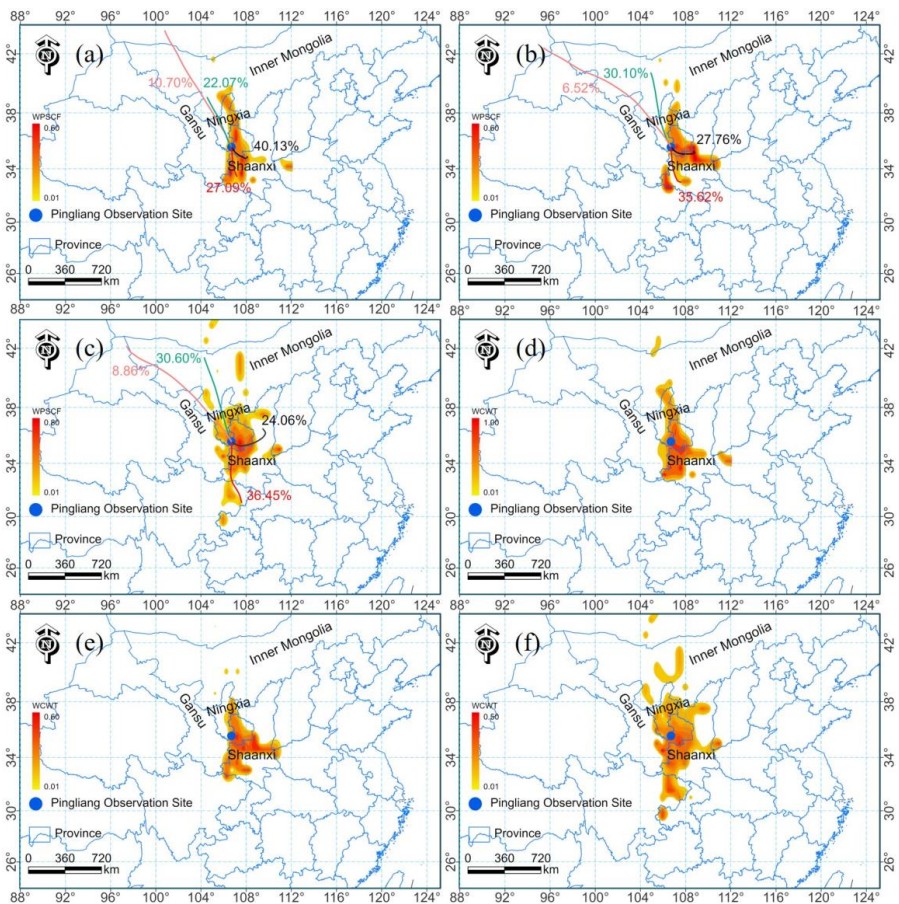

Figure 6. Backward trajectory clustering and PSCF analysis results of air masses at heights of

a) 100, b) 500, and c) 1000 m during the observation period, respectively. CWT analysis

results of air masses at heights of d) 100, e) 500, and f) 1000 m during the observation period,

respectively.

3.3.2 Impacts of local sources on carbonaceous aerosols

In addition to long-range transport, local wind speed and direction significantly

modulate the vertical distribution of aerosols at the observation site. Conditional

Probability Function (CPF) plots, which relate pollutant concentrations to wind

sectors and speeds, help elucidate these local transport and dispersion mechanisms.

Figure 7 shows CPF diagrams for the 90th percentile of UVPM concentrations at 100

m, 500 m and 1000 m. The mean UVPM concentrations decrease with altitude, from



0.64 µg m$^{-3}$ at 100 m, to 0.44 µg m$^{-3}$ at 500 m, and to 0.31 µg m$^{-3}$ at 1000 m,
indicating that extreme UVPM events become less probable aloft. At 100 m, elevated
UVPM levels are associated with southerly, southwesterly, northerly and
northwesterly winds, reflecting both urban emissions and regional inflow. At 500 m,
the influence of northwesterly, southwesterly and northerly sectors diminishes, while
southerly winds remain the dominant driver of high UVPM, albeit with reduced
effect. Southeasterly winds sporadically produce high UVPM at both 100 m and 500
m, likely linked to local point sources, but occur infrequently. Overall, southerly and
southeasterly sectors exert the greatest influence on UVPM at all levels, consistent
with the location of Pingliang's urban center directly south of the measurement site
(Figure 1).

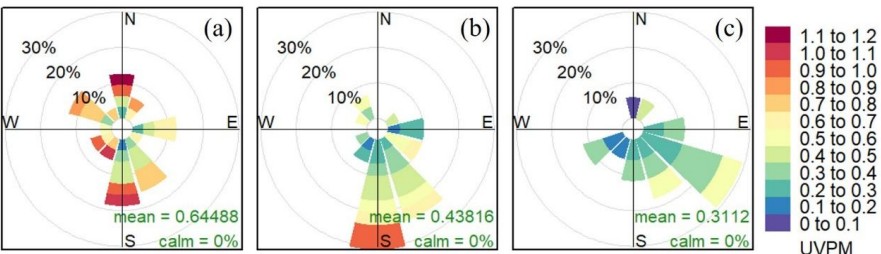


Figure 7. CPF (Conditional Probability Function) diagrams of the 90th percentile of the
UVPM concentration with respect to wind speed and wind direction at the heights of a) 100,
b) 500 and c) 1000 m.

To better compare the thermodynamically driven and dynamically driven influences
on pollutant vertical distribution, this section again focuses on 27 July 2024 to
examine the effects of wind speed and mechanical turbulence. Figure 8 presents
horizontal and vertical wind speeds together with mechanical turbulence measured by
the Doppler wind lidar during the sampling periods on that day. At 05:00 LST, both
horizontal and vertical wind speeds within the boundary layer were low, while
mechanical turbulence was comparatively high, promoting efficient vertical mixing
and producing a nearly uniform distribution of carbonaceous aerosol concentrations
throughout the boundary layer. In the residual layer above, stronger vertical winds





combined with weaker turbulence transported aerosols upward, causing UVPM and
BC concentrations to increase with height between 400 and 600 m. At 08:00 LST,
residual-layer concentrations exhibited a pattern similar to that at 05:00, but stronger
vertical winds and reduced turbulence at the residual-layer top led to a more
pronounced concentration decrease above the layer. By late morning, enhanced
vertical wind speeds within the boundary layer facilitated upward transport of
pollutants. However, at both 11:00 and 14:00 LST the aerosol concentration profile
remained relatively uniform, with only a slight decrease with height; this reflects the
competing effects of thermal convection, which lifts near-surface pollutants, and
elevated turbulence aloft, which strengthens vertical exchange. At 17:00 LST,
subsiding motions prevailed at 900–1200 m and mechanical turbulence index
throughout the column was low, weakening vertical exchange and causing pollutants
to accumulate near 800 m. By 20:00 LST, weak mechanical turbulence at the
boundary-layer top and widespread subsidence near the surface suppressed upward
transport of carbonaceous aerosols; as a result, pollutant concentrations decreased
sharply with height within the boundary layer, while above the boundary-layer top
horizontal and vertical winds aided dispersion, producing a continued decline in
aerosol concentrations up to 800 m.

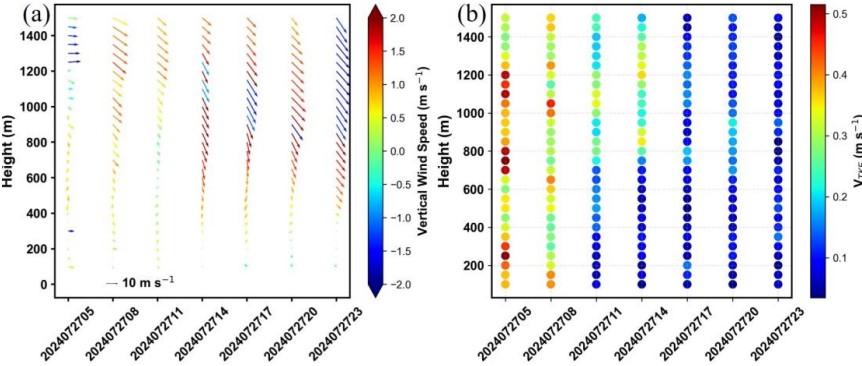


Figure 8. a) Vertical wind speed, horizontal wind speed and direction at 100–1500 m at the
varying hours on July 27, 2024. The arrows indicate horizontal wind direction, with arrows
pointing upward representing northerly winds. Arrow length represents horizontal wind speed
magnitude. The color-shaded areas represent vertical wind speed. b) The $V_{TKE}$ values at





varying altitudes, with distinct colors representing different $V_{TKE}$ values.

To systematically investigate the mechanisms by which thermodynamic and dynamic
processes influence the vertical distribution of carbonaceous aerosols, the observation
periods were grouped into daytime (8:00, 11:00, 14:00 and 17:00 LST) and nighttime
(20:00, 23:00 and 5:00 LST). A random forest nonlinear regression was applied to
quantify the relative contributions of thermal forcing and dynamic forcing to the
vertical concentration gradients of carbonaceous aerosols at different altitude layers
(Figure 9). All regressions achieved coefficients of determination above 0.70,
indicating good explanatory power. During daytime, these two processes exhibit clear
vertical stratification. From the surface up to 600 m, thermal forcing dominates the
evolution of aerosol concentration, whereas between 600 m and 1000 m horizontal
wind speed is the primary driver. At night, the influence of thermal forcing is more
complex. Between the surface and 300 m both thermal forcing and horizontal wind
speed jointly govern vertical concentration variability. Between 300 m and 500 m,
thermal forcing alone exerts decisive control, while above 500 m dynamic processes
exert a much stronger influence than thermodynamic processes. Comparison with
boundary layer height analyses shows that 300 m corresponds to the inversion top at
night, where both thermodynamic and dynamic mechanisms contribute comparably to
aerosol pollution. The layer from 300 m to 500 m largely coincides with the residual
layer, which retains daytime turbulence characteristics and therefore responds more
sensitively to thermal forcing. It should be noted that daytime measurements were
taken at 08:00, 11:00 and 17:00 LST, a period when the boundary layer had not yet
fully developed, so that 600 m approximately corresponds to the daytime boundary
layer top. Consequently, within the daytime boundary layer, carbonaceous aerosol
vertical distributions are mainly controlled by thermodynamic processes, in contrast to
the thermodynamic dominance within the nocturnal residual layer. A schematic of
these regulatory mechanisms for aerosol vertical structure is presented in Figure 10.

Li et al. (2019) also found through radiosonde observations that during the daytime,



thermodynamic processes induced unstable stratification within the boundary layer,
resulting in well-mixed aerosols. In contrast, at night, the stable atmospheric
stratification from the surface to 200 meters suppressed vertical dispersion of
aerosols. Between 500 and 1000 meters, the presence of a low-level jet significantly
influenced the vertical distribution of aerosols. In addition, strong mechanical
turbulence played a key role in facilitating aerosol dispersion near the top of the
boundary layer (Sun et al., 2024). These findings are consistent with the conclusions
of this study.

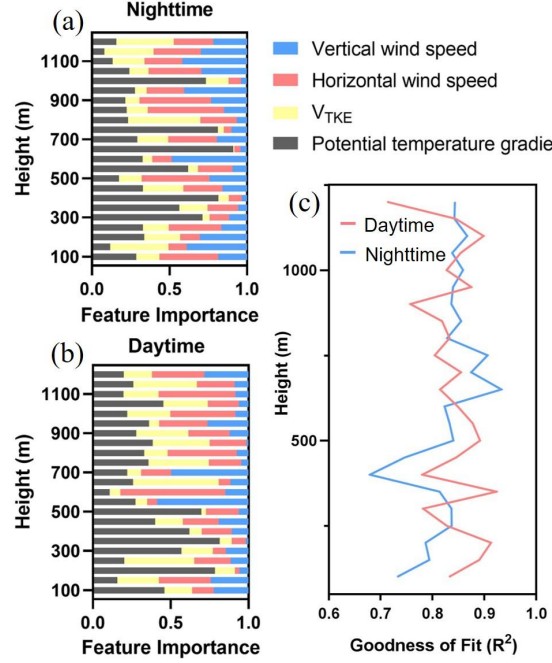


Figure 9. Results of feature importance analysis for the impacts of potential temperature
gradient, mechanical turbulence, horizontal wind speed, and vertical wind speed on the
UVPM gradient during a) nighttime and b) daytime, respectively. c) model goodness of fit
($R^2$) in the calculation.



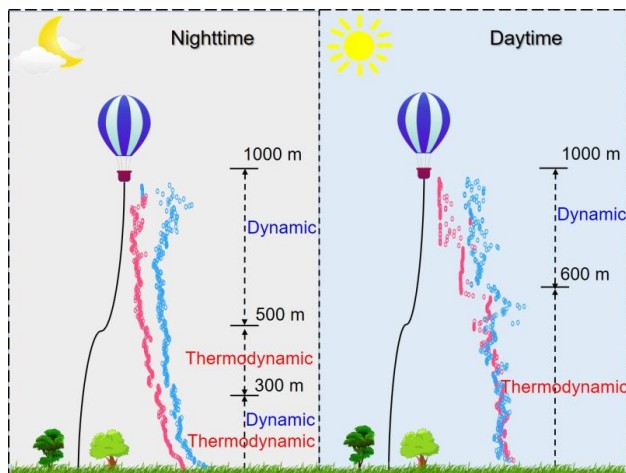

Figure 10. Schematic illustration of thermodynamic and dynamic impacts on aerosol vertical distribution. Within the figure, red circles denote IR BC concentrations, whereas blue circles denote UVPM concentrations; the more rightward a circle's position, the higher the corresponding concentration.

## 4. Conclusions

Pingliang City is situated on the Loess Plateau in northwestern China, where observational data on the vertical distribution of carbonaceous aerosols and meteorological parameters within the planetary boundary layer remain limited. To address this data gap, this study conducted detailed vertical profiling in a typical tableland region of the Loess Plateau using tethered balloons equipped with relevant observation instruments during July 2023 and July 2024.

The study found that near-surface concentrations of black carbon (BC) and ultraviolet-absorbing particulate matter (UVPM) in Pingliang were 0.10 μg m$^{-3}$ and 0.12 μg m$^{-3}$, respectively. These concentrations are slightly lower than those reported for major Chinese cities including Beijing, Shanghai, Nanjing, Chengdu, Shenzhen, Hengshui, the Beibu Gulf region, and Lanzhou, as well as European cities such as Stuttgart in Germany and Milan in Italy. However, they are higher than the BC levels





observed over the Tibetan Plateau and in the Arctic. A comparison of the vertical
profiles of BC and UVPM showed that during early morning and nighttime periods,
when convective activity is relatively weak, UVPM concentrations in the upper
atmosphere are generally higher than those of BC. Near the surface, the difference
between BC and UVPM concentrations is relatively small. This phenomenon is likely
related to the formation of new particles in the upper atmosphere through gas to
particle conversion of gaseous pollutants.

Analysis of thermodynamic and dynamic processes influencing the vertical
distribution of carbonaceous aerosols shows that thermodynamic processes primarily
govern vertical transport in the near-surface layer, while enhanced dynamic processes
in the upper atmosphere promote horizontal dispersion of pollutants. The influence of
thermodynamic and dynamic mechanisms on aerosol vertical profiles exhibits distinct
stratification between daytime and nighttime. At various altitudes, air masses
originating from the south are consistently associated with elevated UVPM
concentrations. This pattern may be attributed to the combined influence of pollution
sources located in the urban area to the south of the site and topographic differences
along the north and south directions.

Nevertheless, there are still some limitations in this study that should be addressed in
future work. Firstly, observations under strong wind conditions in the upper air were
not successfully conducted during the campaign. Secondly, the study primarily
focused on a limited set of air pollutants. Future research will incorporate additional
gaseous pollutants such as $SO_2$, $NO_2$, $O_3$, and VOCs to enable a more comprehensive
analysis of the chemical formation mechanisms and vertical distribution
characteristics of aerosols. Furthermore, the filed campaign was conducted at only a
site, and thus the feedbacks between aerosols and PBL meteorology cannot be fully
understood at whole Loess Plateau. The upcoming field campaign will be
conducted at the other sites to better reveal the impact of thermodynamic and dynamic
processes on the vertical profiles of air pollutants.




**Author contributions.** QS performed the data analysis and prepared the initial draft
of the manuscript. SZ and YY designed the experimental approach and revised the
manuscript. QS, SZ, LD, TZ, GZ, JL, XZ, and YL participated in data collection
during the experiment.

**Financial support.** This work was supported by the National Natural Science
Foundation of China (42422504), Major Science and Technology Project of Gansu
Province (24ZD13FA003), and Excellent Member of Youth Innovation Promotion
Association, Chinese Academy of Sciences (Y2021111), and Youth United Funding of
Lanzhou Branch of Chinese Academy of Sciences.

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
