# Peer review of "Impacts of Thermodynamic and Dynamic Processes on the Vertical"

_EGUsphere, 2025_

## Referee Comment (RC1)

**Referee Comments – Impacts of Thermodynamic and Dynamic Processes on the Vertical Distribution of Carbonaceous Aerosols: lessons from in-situ observations at eastern foothills of LiuPan Mountains, Loess Plateau**

(https://doi.org/10.5194/egusphere-2025-3254)

**General Overview:**

The manuscript (egusphere-2025-3254) presents an interesting study investigating the vertical distribution of light-absorbing aerosols based on in-situ observations over Liupan Mountains in China. The topic of this study falls within the scope of the journal Atmospheric Chemistry and Physics (ACP). The authors present data from field measurements and analyze the impacts of thermodynamic and dynamic processes on the vertical distribution of light-absorbing aerosols. The authors provide insights and helpful information for a better understanding of light-absorbing aerosols in the Loess Plateau of China and potentially other places with similar atmospheric environments. This manuscript is generally laid out well and shows its academic value. This manuscript is recommended to be published after addressing the concerns and comments below with minor revisions.

**Major Concern:**

- The definitions of the terms "UVPM", "BC", and "IRBC" in this study are not clear and thus might influence the interpretation of data and cause confusion for the readers. The terms are also used inconsistently throughout the manuscript, raising concerns of logical conflict in the study. In Line 20, UVPM seems to be the acronym for "ultraviolet particulate matter" while the other term "ultraviolet absorbing particulate matter" also appears to be the full form of UVPM in Line 605. More details are described as separate bullet points listed below.

- Lines 159 – 161: Is black carbon (BC) concentration measured at wavelength 880 nm denoted as BC or IRBC? From the context of the whole manuscript, it seems that BC represents the particle concentration measured at wavelengths 470 nm, 528 nm, 625 nm, and 880 nm, while IRBC represents only the particle concentration measured at wavelength 880 nm, and the particle concentration measured at wavelength 375 is denoted as UVPM. Please ensure consistency throughout the manuscript and avoid confusion for the readers.

- Lines 161 – 163: Where does this definition of UVPM come from? Please provide relevant citations or explicitly mention that it is specifically defined by this study as

the term "ultraviolet particulate matter" and "ultraviolet absorbing particulate matter" are not yet a standardized term in the literature. Please also provide the reason for using this term to convince the readers if using this term is necessary to be created as opposed to using the more commonly used brown carbon or dust aerosol when it comes to absorption. Based on the context of the whole manuscript, the term UVPM seems to be the total absorption contributed from dust, brown carbon, and black carbon at the wavelength 375 nm, but it would be great if this could be clarified by the authors.

- Lines 176 – 178: If only the absorption at wavelength 880 nm is considered as black carbon according to the statement in the previous section (Lines 159 – 161), why are the absorptions at multiple wavelengths (470 nm, 528 nm, and 625 nm) used for the calculation of light absorption coefficient of black carbon? Please revise the corresponding statements to avoid conflicts of logic throughout the manuscript.

**Minor Concerns:**

- Lines 156 – 158: Does this instrument "MicroAeth® MA350" attribute all light absorption to black carbon? How do the authors quantitatively distinguish black carbon from other light-absorbing aerosols? What is the corresponding uncertainty for this method to quantify black carbon?

- Line 183: Please cite the source of Equation 2.

- Line 200: Please cite the source of Equation 4.

- Line 203: Please cite the source of Equation 5.

- Lines 207 – 208: The definition of the term "planetary boundary layer (PBL)" should be provided when the term first appears in the text.

- Line 213: Please provide citations to support this statement "various methodologies exist for determining PBL height (PBLH)" or revise the text by simply describing the approaches used in this study. It is recommended to provide citations while stating that various methodologies exist.

- Line 227: Please cite the source of Equation 6.

- Lines 245 – 248: It would be great if this qualitative statement can be turned into a table for a quantitative comparison, which would make the argument more convincing and make this work more valuable and academically citable.

- Lines 253 – 254: Without showing the quantitative data for direct comparisons of emission inventories of air pollutants from daily human activities, it would be safer to use a more conservative statement like "probably lower" to ensure scientific rigor and avoid controversy.

- Line 288: It seems that only diurnal variations of IRBC rather than BC are shown.

- Line 290: Are the profiles for BC or IRBC?

- Line 295: The term IRBC is used here while referring to BC mentioned earlier (Line 290). It seems that IRBC only represents black carbon (BC) measured at wavelength 880 nm while the absorption measured at wavelengths 470nm, 528nm, 625 nm, and 880 nm are all considered as black carbon in this study. If this is the case, please revise the definition in earlier sections of this manuscript for consistency.

- Line 300: Are the profiles for BC or IRBC?

- Line 304: Is the BC here meant to be IRBC?

- Line 305: Is the UVPM-to-BC here meant to be UVPM-to-IRBC?

- Line 316 – 317: Could this change due to vertical mixing and dominance of black carbon? The disparity between UVPM and IRBC should not disappear at lower altitudes if the pollutants considered as UVPM mixed downward during the convection do not include black carbon. If UVPM is considered as the total absorption of dust, brown carbon, and black carbon, this reduction of disparity might indicate that the contribution of dust and organics (brown carbon) becomes less important, and IRBC is dominant during this period.

- Lines 321 – 324: This observation also indicates that the UVPM defined in this study is probably the overall contribution of black carbon, brown carbon, and dust aerosol. Please clarify the definition.

- The term "LT" is used in Lines 225 – 226 while the term "LST" is used in other parts of the manuscript. Please explicitly distinguish the difference between the two terms. Otherwise, please choose one and stick to it to ensure consistency and avoid confusion.

- Figure 3: It seems that IRBC rather than BC is shown in the figure. Please update the caption and revise relevant statements in the main text.

- Section 3.1.3: It seems that the BC used in this section should be IRBC. Please confirm.

- Lines 364 – 366: How are the weather conditions defined based on what parameters and what data? Please provide relevant information.

- Lines 375 – 377: Please provide citations of previous studies.

- Lines 407 – 408 and Lines 414 – 416: The potential-temperature gradient method is stated to yield accurate estimates at night and in the early morning in Lines 407 – 408. However, the potential-temperature gradient method is stated to overshoot the inversion top in Lines 414 – 416, which seems to be contradictory with the earlier statement of accurate estimates. Please revise the statements to clarify and avoid confusion.

- Figure 5: Why are the PBLH (Parcel) lines only shown in some of the panels in the figure while the PBLH (Gradient) lines are shown in all panels? Please clarify in the revision.

- Lines 504 – 505: Point sources have continuous emissions. How can they be linked to high concentrations only infrequently or attributed to the cause of infrequently occurred high concentrations under the same wind direction? Please provide supporting information or data to strengthen this statement.

- Lines 567 – 568: What about 14:00?

- Lines 570 – 572: Please double check the logic of this statement. It seems to be meant to contrast between thermodynamic and dynamic processes while the thermodynamic process is mentioned twice here.

- Figure S2: Is there a specific definition of the term "conventional air pollutants"? It would be great if the caption could be specifically listing the air pollutants shown in this figure.

- Figures S3 and S4: The terms "ascent" and "descent" are used in the main text and figure caption while the terms "rising" and "landing" are used in the figure legend, which are not consistent with each other. It would be great if the terminology could be consistent throughout the work.

**Technical Comments:**

- Line 23: The parameter "VTKE" seems to be a typographical error of "$V_{TKE}$".

- Line 413: The word "and" in front of the word "often" seems to be a typographical error and should be deleted.

- Line 637: The word "understanded" seems to be a typographical error of "understood".

---

## Author Comment (AC1)

**Response to Comments from Reviewer**

**Journal:** Atmospheric Chemistry and Physics

**Manuscript ID:** egusphere-2025-3254

**Title:** " Measurement report: Measurement report: Impacts of Thermodynamic and Dynamic Processes on the Vertical Distribution of Carbonaceous Aerosols: lessons from in-situ observations at eastern foothills of LiuPan Mountains, Loess Plateau"

**Author(s):** Shaofeng Qi, Suping Zhao*, Ye Yu, Longxiang Dong, Tong Zhang, Guo Zhao, Jianglin Li, Xiang Zhang, and Yiting Lv

*Correspondence to*: Suping Zhao (zhaosp@lzb.ac.cn)

In the response, the content in **black** font represents the original text from the manuscript, the content in **blue** font indicates additions or revisions, and the content in **red** font is the response to the reviewers' questions and suggestions.

**Dear Editors and Reviewers:**

Thank you for your and the reviewers' comments concerning our manuscript entitled "Impacts of Thermodynamic and Dynamic Processes on the Vertical Distribution of Carbonaceous Aerosols: lessons from in-situ observations at eastern foothills of LiuPan Mountains, Loess Plateau" (https://doi.org/10.5194/egusphere-2025-3254). Those comments are all very valuable, greatly assisting us in revising and improving our paper, as well as providing important guidance for our research. We have carefully studied the comments and made corrections accordingly, which can be viewed in the revised version of our manuscript. The main corrections in the paper and the responses to the reviewer's comments are provided below:

RESPONDS TO COMMENTS FROM REVIEWER #2

First of all, we appreciate your very positive evaluation of our work. The responses of your specific comments/questions are outlined in detail below.

**Major comments:**

1. The carbonaceous aerosol mass concentration measurements are questionable. There are a few major issues as follows:

1.1 MicroAeth MA350 is a light absorption measurement instrument. It does not directly measure the mass concentration of BC and BrC, but the equivalent BC (eBC) mass concentration. It measures the light absorption coefficients and uses predefined MAC at each wavelength to estimate the mass concentration of BC and BrC. Due to the highly variable MAC for BC and BrC, the predefined MAC values can lead to significant uncertainties in the mass concentration of light-absorbing aerosols. Thus, the mass concentration reported by MA350 should be used qualitatively, and eBC should be used instead of BC in the manuscript.

**Response:** Thank you for this valuable suggestion. We have accordingly revised the description of the MA350 instrument in Section 2.1.2 of the manuscript. In addition, for the data measured at 375 nm (UV), based on the instrument specifications (https://aethlabs.com/products/ma300, last accessed 13 November 2025) and the study by Zhao et al. (2023), these measurements are defined as ultraviolet-absorbing particulate matter (UVPM), which includes brown carbon and black carbon. Furthermore, the measurements at 880 nm (IR) have been defined as equivalent black carbon (eBC), and all references to "IR BC" have been consistently updated to "eBC" throughout the manuscript.

Reference

Zhao, S. P., He, J. J., Dong, L. X., Qi, S. F., Yin, D. Y., Chen, J. B., Yu, Y. Contrasting Vertical Circulation between Severe and Light Air Pollution inside a Deep Basin Results from the Collaborative Experiment of 3D Boundary-Layer Meteorology and Pollution at the Sichuan Basin (BLMP-SCB). Bull. Am. Meteorol. Soc., 104(2), E411-E434, http://doi.org/10.1175/BAMS-D-22-0150.1, 2023a.

1.2 When you calculate the UVPM mass, did you subtract eBC mass?

**Response:** Thank you for your question and suggestion. In this manuscript, we have updated the definitions of the substances measured at 375 nm and 880 nm. The details are as follows:

2.1.2 Carbonaceous aerosol mass concentrations

The MicroAeth® MA350 determines carbonaceous aerosols concentration based

on the Beer-Lambert law, quantifying light absorption by carbonaceous particles deposited on a PTFE filter at multiple wavelengths. The instrument employs five laser wavelengths: 375 nm (UV), 470 nm (Blue), 528 nm (Green), 625 nm (Red), and 880 nm (IR). Carbon concentrations measured at 880 nm are considered to represent the equivalent black carbon concentration (denoted as eBC) (Zhao et al., 2023a), whereas those measured at 375 nm correspond to ultraviolet-absorbing particulate matter (UVPM), which includes primary brown carbon and black carbon emitted from sources such as biomass burning and coal combustion, as well as secondary brown carbon formed through atmospheric oxidation processes (The relevant introduction/details regarding the substance measured at a wavelength of 375 nm can be referenced in the official manual of the MicroAeth® MA Series MA300 instrument. URL: https://aethlabs.com/products/ma300, last accessed: 13 November 2025).

Reference

Zhao, S. P., He, J. J., Dong, L. X., Qi, S. F., Yin, D. Y., Chen, J. B., Yu, Y. Contrasting Vertical Circulation between Severe and Light Air Pollution inside a Deep Basin Results from the Collaborative Experiment of 3D Boundary-Layer Meteorology and Pollution at the Sichuan Basin (BLMP-SCB). Bull. Am. Meteorol. Soc., 104(2), E411-E434, http://doi.org/10.1175/BAMS-D-22-0150.1, 2023a.

1.3 Did you add a diffusion dryer in front of your MA350? If not, your data at high RH should be excluded since high RH can bias filter-based optical measurements.

**Response:** Thank you for your comment. In Fig. S1, we plotted the humidity recorded by the MA350 for the samples. It can be seen that during the observation period, the sample humidity did not exceed 60%, and we consider the measurements to be reliable under these conditions.

[Figure]

Figure S1. Temporal variations of meteorological parameters, including air temperature (T), relative humidity (RH), wind direction (WD), wind speed (WS), and air pollutants, including particulate matter with aerodynamic diameter smaller than 2.5 μm ($PM_{2.5}$) and than10 μm ($PM_{10}$), nitrogen dioxide ($NO_2$), ozone ($O_3$), sulfur dioxide ($SO_2$), and carbon monoxide (CO), were observed during the monitoring period. The shaded areas in the figure represent the during and after vehicle emissions, no dust pollution, and dust pollution at 20:00 on July 17, 23:00 on July 17, 2024, 08:00 on July 20, 2024, 08:00 on July 21, 2024, respectively.

1.4 Did you apply any corrections to your results?

**Response:** Thank you for your question. In this study, the MA350 data were pre-processed using the Optimized Noise-reduction Averaging (ONA) method (as already described in Section 2.1.2). No other data were modified or corrected. The details are as follows:

Negative optical attenuation (ATN) values occasionally occurred under low aerosol concentrations at high altitudes, which were corrected using the Optimized Noise-reduction Averaging (ONA) algorithm (Hagler et al., 2011; Guan et al., 2022).

2. Your source apportionment part is also very confusing. You need to provide more details (e.g., how long was your back trajectory, what is each line in Fig. 9? Are there any altitudes or times? What is the end time? How did you calculate PSCF and CWT? How did you get the source information). All of this information should be provided.

**Response:** Thank you for your suggestion. This will certainly enhance the readability of our study. We have added the relevant details to the manuscript.

We have now added Text 1 to the Supplementary Information, which provides detailed information on the back trajectory duration, start and end times, and the methodologies used to calculate WPSCF and WCWT.

Additionally, the reviewer's query regarding "each line in Fig. 10" is likely referring to Fig. S10. Fig. S10 illustrates all air mass trajectories that reached the observation site at 500 m altitude during the observation campaign, where different colors represent the change in altitude of these air masses during their transit. To improve the clarity and readability of Fig. S10, we have updated the figure caption with additional explanatory details.

The specific revisions are as follows:

(1) Text 1 in the Supplementary Material

**Text 1: Analysis methods of WPSCF and WCWT**

In the calculation of WPSCF (weighted potential source contribution function) and WCWT (weighted concentration weighted trajectory), we first computed the air mass trajectories for the preceding 48 hours at heights of 100 m, 500 m, and 1000 m within all observation days (July 15–24, 2023, and July 17–31, 2024). Next, the UVPM mass concentration data were overlaid onto the air mass trajectory data, and the areas traversed by the air mass trajectories were divided into a 0.5°×0.5° grid, in order to determine the potential source regions of UVPM for the observation station. The calculation methods for WPSCF and WCWT are shown in Equations (1) and (2) (Wang, 2014):

$$\text{WPSCF}_{ij} = \frac{m_{ij}}{n_{ij}} \times W_{ij} \tag{1},$$

$$C_{ij} = \frac{1}{\sum_{l=1}^{M} \tau_{ijl}} \times \sum_{l=1}^{M} C_l \tau_{ijl} \tag{2}.$$

In Equation (1), $m_{ij}$ represents the number of polluted trajectory endpoints within the ij-th cell, and $n_{ij}$ represents the total number of trajectory endpoints within the ij-th cell. $W_{ij}$ is the weighting factor. In Equation (2), $C_{ij}$ is the average weighted concentration

in the ij-th cell, l is the index of the trajectory, M is the total number of trajectories, $C_l$ is the concentration observed on arrival of trajectory l, and $\tau_{ijl}$ is the time spent in the ij-th cell by trajectory l. A high value for $C_{ij}$ implies that air parcels traveling over the ij-th cell would be, on average, associated with high concentrations at the receptor. The arbitrary weighting function described above was also used in the CWT analyses to reduce the effect of the small values of $n_{ij}$. (All the above content refers to the official MeteoInfo documentation, which can be found at the following URL: http://www.meteothink.org/docs/trajstat/trajstatrun.html.)

(2) Revision of the Fig. S10 caption.

[Figure]

Figure S10. All 48-hour backward trajectories of air masses that reached the Pingliang Observatory at 500 m height (above ground level) during the measurement period (from July 15 to 24, 2023, and from July 17 to 31, 2024), where the different colors of each trajectory represent the altitude of the air mass at different geographic locations.

3. You need to provide more details about your TBS flights. What was your flight pattern? You should show your results as altitude vs times (See an example as Figure 2 in "Vertical Gradient of Size-Resolved Aerosol Compositions over the Arctic Reveals Cloud Processed Aerosol in-Cloud and above Cloud"). It is very difficult for me to understand your results without seeing your flight pattern.

**Response:** Thank you for your suggestion. We have added the relevant statistics for each observation in the supplementary material. Figure S2a shows the number of observations during different hours of day, while Figures S2b and S2c present the observation heights for each measurement and the eBC and UVPM concentrations at

[Figure]

Figure S2. a) Statistics of the number of observations for different hours of day, while b) and c) the observation heights for each measurement and the eBC and UVPM concentrations at different heights, respectively.

4. Since the number of your measurements is still within a reasonable amount, I suggest adding individual flight results to SI.

**Response:** Thank you for your suggestion. We have added the relevant statistics for each observation in the supplementary material. Figure S2a shows the number of observations during different hours of day, while Figures S2b and S2c present the

observation heights for each measurement and the eBC and UVPM concentrations at different heights, respectively.

[Figure]

Figure S2. a) Statistics of the number of observations for different hours of day, while b) and c) the observation heights for each measurement and the eBC and UVPM concentrations at different heights, respectively.

5. Could you provide details about how you combined and average flights that started from the same hours?

**Response:** Thank you for raising this question. In this study, we interpolated the eBC and UVPM data for each observational profile within its maximum measurement height

and then calculated the mean values across all profiles obtained during the same observational period. To avoid errors associated with extrapolation during the interpolation process, data beyond the maximum observed height of each profile were assigned as missing values, and these missing values were excluded from the averaging procedure.

6. It is not clear to me what threshold you used for PBL height, and why you got different PBL heights for the two methods, and which one I should rely on.

**Response:** Due to adjustments made to the figures within the manuscript, this illustration has been renumbered as "Figure 6". Due to the discrepancies in results from different methods for calculating the thermal PBL, two methods were selected in this study to calculate the PBL height for daytime and nighttime, respectively. Since Holzworth (1964) defined the Parcel Method (Maximum Mixing Depth, MMD) as: "The height at which a dry adiabatic line extending upward from the maximum surface temperature intersects the most recently observed temperature profile," it is suitable for unstable atmospheric conditions and provides more accurate results for the daytime PBLH calculation. The nighttime PBLH, however, is calculated using the potential temperature gradient. This explanation can be found in the Section 2.3 and first paragraph of Section 3.2 of the manuscript. In addition, we revised Figure 6 by retaining only the PBLH derived from the observational profiles. The specific content is as follows:

In Section 2.3:

Planetary boundary layer is the part of the atmosphere closest to the planet's surface, accounting for approximately 10% – 20% of the troposphere. It is the lowest layer of the troposphere directly influenced by surface forcing, with a response time of less than one hour, playing a critical role in the dispersion and transport of air pollutants. Within the PBL, turbulent mixing processes homogenize air temperature and humidity, resulting in relatively uniform distributions of these properties. PBL height (PBLH) refers to the thickness of the layer most significantly affected by the surface. In other

words, it is the vertical extent that surface turbulent motion (caused by surface heating, friction, or topography, etc.) can effectively influence (Emeis et al., 2008; Seibert et al., 2000). This study employs two established methodologies for determining the PBLH: the potential temperature gradient method and the parcel method (Holzworth, 1964; Zhang et al., 2020). The potential temperature gradient method was utilized for calculating the PBLH during the nighttime and early morning hours (20:00, 23:00, 05:00, and 08:00 LT), while the parcel method was applied specifically for the daytime periods (11:00, 14:00, and 17:00 LT). Detailed computational procedures for both approaches are summarized in Supplementary Table S1.

In Section 3.2:

The parcel method is well suited to convective conditions but cannot accurately resolve PBLH during early morning and nighttime (Holzworth, 1964), whereas it performs reliably under strong daytime convection. Using this approach, PBLH at 11:00 and 17:00 LT were 503 m and 459 m, respectively; at 14:00, measurement ceilings did not reach PBLH. By contrast, the potential temperature gradient method yields accurate estimates at night and in the early morning: PBLH at 05:00, 08:00, 20:00, and 23:00 LT were 255 m, 181 m, 260 m, and 216 m, respectively.

[Figure]

Figure 6. Diurnal variations in profiles of eBC, UVPM and potential temperature on July 27,

2024. The red line represents eBC, the blue line represents UVPM, and the green line represents the potential temperature profile. The light orange shaded area represents the difference between UVPM and eBC. The black dashed lines represent the PBLH.

References

Holzworth, G. C. Estimates of mean maximum mixing depths in the contiguous united states. Mon. Weather Rev., 92(5), 235-242, http://doi.org/10.1175/ 1520-0493(1964)092<0235:EOMMMD>2.3.CO;2, 1964.

Zhang, H. S., Zhang, X. Y., Li, Q. H., Cai, X. H., Fan, S. J., Song, Y., Hu, F., Che, H. Z., Quan, J. N., Kang, L., Zhu, T. Research progress on estimation of atmospheric boundary layer height. Acta Meteorol. Sin., 78(3), 522-536, http://doi.org/10.11676/qxxb2020.044, 2020.

**General comments:**

1. Your Introduction misses recent TBS work from the Atmospheric Radiation Measurement (ARM).

**Response:** Thank you for your valuable suggestions. We have now cited those important studies in the main text, and the revised content is as follows:

In contrast, tethered-balloon systems mitigate these key constraints by providing high-resolution BC profiles within the PBL. This approach has been widely applied in various regions across the globe, including Europe, the Arctic, the United States, and South Asia, as well as several key regions in China, such as the North China Plain, the Sichuan Basin, and the Yangtze River Delta (Bisht et al., 2016; Mazzola et al., 2016; Guagenti et al., 2025; Ran et al., 2016; Samad et al., 2020; Wang et al., 2018b; Zhao et al., 2023a).

References

Bisht, D. S., Tiwari, S., Dumka, U. C., Srivastava, A. K., Safai, P. D., Ghude, S. D., Chate, D. M., Rao, P. S. P., Ali, K., Prabhakaran, T., Panickar, A. S., Soni, V. K., Attri, S. D., Tunved, P., Chakrabarty, R. K., Hopke, P. K. Tethered balloon-born and ground-based measurements of black carbon and particulate profiles within the lower troposphere during the foggy period in Delhi, India. Sci. Total Environ., 573, 894-905, http://doi.org/10.1016/j.scitotenv.2016.08.185, 2016.

Guagenti, M. and Dexheimer, D. and Ulinksi, A. and Walter, P. and Flynn III, J. H. and Usenko, S. A modular approach to volatile organic compound samplers for tethered balloon and drone platforms. Atmos. Meas. Tech., 18(9), 2125-2136, https://doi.org/10.5194/amt-18-2125-2025, 2025.

Mazzola, M., Busetto, M., Ferrero, L., Viola, A. P., Cappelletti, D. AGAP: an atmospheric gondola for aerosol profiling. Rendiconti Lincei-Scienze Fisiche E Naturali, 27(Suppl 1), 105-113, http://doi.org/10.1007/s12210-016-0514-x, 2016.

Ran, L., Deng, Z. Z., Xu, X. B., Yan, P., Lin, W. L., Wang, Y., Tian, P., Wang, P. C., Pan, W. L., Lu, D. R. Vertical profiles of black carbon measured by a micro-aethalometer in summer in the North China Plain. Atmos. Chem. and Phys., 16(16), 10441-10454, http://doi.org/10.5194/acp-16-10441-2016, 2016.

Samad, A., Vogt, U., Panta, A., Uprety, D. Vertical distribution of particulate matter, black carbon and ultra-fine particles in Stuttgart, Germany. Atmos. Pollut. Res., 11(8), 1441-1450, http://doi.org/10.1016/j.apr.2020.05.017, 2020.

Wang, Q. Q., Sun, Y. L., Xu, W. Q., Du, W., Zhou, L. B., Tang, G. Q., Chen, C., Cheng, X. L., Zhao, X. J., Ji, D. S., Han, T. T., Wang, Z., Li, J., Wang, Z. F. Vertically resolved characteristics of air pollution during two severe winter haze episodes in urban Beijing, China. Atmos. Chem. Phys., 18(4), 2495-2509, http://doi.org/10.5194/acp-18-2495-2018, 2018b.

Zhao, S. P., He, J. J., Dong, L. X., Qi, S. F., Yin, D. Y., Chen, J. B., Yu, Y. Contrasting Vertical Circulation between Severe and Light Air Pollution inside a Deep Basin Results from the Collaborative Experiment of 3D Boundary-Layer Meteorology and Pollution at the Sichuan Basin (BLMP-SCB). Bull. Am. Meteorol. Soc., 104(2), E411-E434, http://doi.org/10.1175/BAMS-D-22-0150.1, 2023a.

2. L40-L42, "OC encompasses … (VOCs). " POC can come from other sources, like vehicle emissions, and SOC can be formed from dark aging.

**Response:** Thank you for drawing our attention to this issue. We have revised this part of the content in the manuscript. The revised content is as follows:

Organic carbon (OC) can be broadly classified into primary organic carbon (POC) and

secondary organic carbon (SOC). The primary organic carbon originates directly from sources such as biomass burning, vehicle exhaust, and industrial emissions, whereas SOC is formed through the oxidation of volatile organic compounds (VOCs) via photochemical and dark reaction pathways. Due to its complex chemical composition and the strong ultraviolet light absorption of certain organic fractions, which are referred to as brown carbon (BrC), OC introduces substantial uncertainty into climate impact assessments (Kroll et al., 2011).

Reference

Kroll, J., Donahue, N., Jimenez, J. Kessler, S., Canagaratna, M., Wilson, K., Altieri, K., Mazzoleni, L., Wozniak, A., Bluhm, H., Mysak, E., Smith, J., Kolb, C., Worsnop, D. Carbon oxidation state as a metric for describing the chemistry of atmospheric organic aerosol. *Nature Chem.*, 133–139, https://doi.org/10.1038/nchem.948, 2011.

3. AAE is Absorption Angstrom Exponent, not Angstrom Absorption Exponent.

**Response:** Thank you for pointing out this error. We have modified this content in the manuscript. The revised content is as follows:

$$b_{Abs}(\lambda) = k \times \lambda^{-AAE} \qquad (2).$$

In Eq. (2), k is a wavelength-independent constant, and AAE represents the Absorption Ångström Exponent, which characterizes the wavelength dependence of light absorption of carbonaceous aerosols (Ångström, 1929). A higher AAE value signifies that the aerosol absorption capacity decreases more rapidly with increasing wavelength. By combining the $b_{Abs}$ values derived from Eq. (1) with Eq. (2), vertical profiles of AAE for carbonaceous aerosols were obtained.

Reference

Ångström, A., 1929. On the atmospheric transmission of sun radiation and on dust in the air. Geogr. Ann. 11, 156–166.

4. Figures S3 and S4, and Figure 2. There are many altitudes showing 0 μg m$^{-3}$ eBC. Could you explain why?

**Response:** Thank you for your question. This is because the equivalent black carbon (eBC) mass concentration in the upper atmosphere is very low, and when measured by optical instruments, the results often yield negative values. When the Optimized Noise-reduction Averaging (ONA) method is used for data quality control, these values are corrected based on the optical attenuation signal, but the resulting values are not zero; instead, they are values close to zero. To improve the readability of the manuscript, we have added a relevant explanation in the text. The revised content is as follows:

Figure 2 shows the averaged profiles for all sampling periods during the observation campaign, the red solid line and its shaded envelope denote the mean and standard deviation of eBC, while the blue solid line and its shaded envelope denote the mean and standard deviation of UVPM. Because eBC mass concentration in the upper atmosphere is extremely low (below the instrument's normal limit of detection), optical measurements often yield negative results, which are corrected when the ONA method is used for data quality control. Furthermore, owing to the relatively high wind speeds in the upper atmosphere, our measurements in the early afternoon were typically unable to reach the PBLH.

5. L290-L292, "The results indicate … mean profile. "It seems to me that only 5 am, 20:00, and 23:00 show acceptable similarities. Thus, I am unsure if you can combine others.

**Response:** Thank you very much for raising this point. We agree with your insightful comment. In this study, we consider that the ascent and descent profiles measured at 5:00, 20:00, and 23:00 exhibit high similarity. Therefore, both ascent and descent profiles at these time points are treated as independent observational profiles for the calculation of mean values, while only the ascent profiles from the other time points are retained for this purpose. We have revised the relevant description in the manuscript, with the specific details presented as follows:

Figures S3 and S4 respectively depict the trends and correlations of the ascent and descent profiles for eBC and UVPM. The results indicate that the ascent and descent profiles at 5:00, 20:00, and 23:00 in this study exhibit similar trends, and thus they can

both be treated as independent observational profiles for analysis. For observations at 8:00, 11:00, 14:00 and 17:00, only the ascent data were used for analysis.

6. Figure 2. I did not see red and blue shaded areas.

**Response:** Thank you for pointing out this issue. This was caused by mistakenly placing a scatter plot of the average values as Figure 2 in the manuscript. We have corrected this in the manuscript, and the revised Figure 2 is shown below:

[Figure]

Figure 2. Averaged diurnal variation in profiles of the eBC and UVPM concentration during the field campaign. The red and blue lines represent eBC and UVPM concentrations, respectively, while the red and blue shaded areas denote the standard deviation of eBC and UVPM, respectively. The light orange shaded area represents the difference between UVPM and eBC.

7. L302-305, "A comparative analysis … smaller differences." Have you done any statistical tests? Why are there no IRBC at high altitudes for 5:00, 11:00, and 20:00? Why do 8:00 and 11:00 not agree with your argument?

**Response:** Thank you for your valuable suggestion.

(1) We have added a description in the main text regarding the pronounced differences between UVPM and eBC at higher altitudes, together with the corresponding observation periods. In addition, we have included a significance test of the UVPM–eBC differences in the supplementary figures. The specific revisions are as follows:

A comparative analysis of their vertical profiles reveals that during periods of weak convective activity, such as early morning and nighttime (i.e., 05:00, 20:00, and 23:00 LT), UVPM concentrations aloft generally exceed those of eBC (the significance test of the UVPM–eBC differences is provided in Figure S5), while near the surface the two species show much smaller differences.

[Figure]

Figure S5. Significance test of the UVPM–eBC differences during different hours of the day. The red and blue dashed lines in the figure indicate p-values of 0.05 and 0.01, respectively.

(2) At 05:00, 11:00, and 20:00, the eBC in high-altitude areas is not zero, but rather close to zero. This may be related to weaker vertical transport in that layer of the atmosphere, where the upward transport of eBC from the lower atmosphere is suppressed.

(3) As shown in Figure S5, at 20:00, 23:00, and 05:00, there are significant differences between UVPM and eBC throughout the entire atmosphere. However, at 08:00 and 11:00, except for small vertical layers where UVPM and eBC show significant differences, there are no significant differences between UVPM and eBC in most of the atmosphere. This may be due to the development of the PBL in the early morning and increased human activities, which lead to higher near-surface pollution levels and upward transport. This causes significant differences between near-surface UVPM and eBC at 08:00. At this time, the PBL is still in the development stage, and enhanced

vertical mixing at the PBL top reduces the difference between UVPM and eBC in the middle and lower atmosphere. Since the PBL has not yet reached the maximum observation height, the significant differences between UVPM and eBC in the upper atmosphere at 08:00 have not disappeared. By 11:00, there are no significant differences between UVPM and eBC in the lower atmosphere.

8. L309-L313, "There, in a … Zhao et al., 2024)." I don't think your data supports this since I did not see much increase in UVPM mass, but I did see more IR BC mass from 20:00-23:00. Moreover, I saw a significant reduction of both IR BC and UVPM at 5:00 compared with 23:00. Please justify this.

**Response:** Thank you for your question and constructive suggestions. Following the method proposed by Wu et al. (2016), we applied a least-squares approach to estimate the secondary component of UVPM (UVPM$_{sec}$) using the ratio between UVPM$_{sec}$ and eBC for each observation sample. We further supplemented the discussion by examining the variations in the UVPM$_{sec}$/eBC ratio across different times and altitudes. The detailed explanation is provided below:

To further investigate the underlying causes, we estimated the concentration of secondary UVPM (UVPM$_{sec}$) using the least-squares method described by Wu et al. (2016), based on the ratio of UVPM to eBC. Figure 3 presents the mean diurnal variations in vertical distributions of the ratios of UVPM$_{sec}$ to total UVPM, and of UVPM$_{sec}$ to eBC. From 17:00 to 05:00 LT, the UVPM$_{sec}$/UVPM and UVPM$_{sec}$/eBC ratios aloft increase markedly, indicating that the elevated UVPM during this period is more strongly influenced by secondary sources. As time progresses, the region characterized by high ratios gradually approaches the ground surface and eventually disappears by 08:00 LT.

The enhanced contribution of secondary sources aloft may be attributed to the increasingly stable stratification after 17:00 LT, which suppresses vertical mixing and inhibits the upward transport of both eBC and UVPM. In contrast, gaseous precursors of UVPM$_{sec}$ (i.e., VOCs) can accumulate above the PBL. Under relatively clean

atmospheric conditions with limited condensation sinks, nocturnal chemistry dominated by $NO_3^-$ radicals, together with low ambient temperatures, favors the formation of secondary organic aerosols through gas-to-particle partitioning and temperature-dependent condensation (Han & Jang, 2023; Kuang et al., 2025; Kulmala, 2022; Morgan et al., 2009; Wang et al., 2023; Zhao et al., 2023b). After sunrise (around 06:00 LT in summer at this site), enhanced solar radiation promotes convective mixing and weakens the nocturnal inversion at the top of the PBL. Consequently, UVPM$_{sec}$-enriched air masses retained within the residual layer are entrained downward into the growing daytime PBL, thereby reducing the concentration gradient between UVPM and eBC within the PBL (Zhao et al., 2023a). In addition, increased human activities and strengthened thermal convection after sunrise lead to larger primary emissions throughout the atmospheric column, causing carbonaceous aerosols to be increasingly dominated by primary components. Because the PBL is still developing at 08:00 LT, the near-surface concentrations of UVPM and eBC are higher than those observed at 05:00 LT. As thermal convection intensifies, the decline in near-surface UVPM and eBC slows between 11:00 and 14:00 LT. Between 14:00 and 17:00 LT, strong convective mixing results in an approximately uniform vertical distribution of both eBC and UVPM from the surface up to ~800 m.

[Figure]

Figure 3. Mean diurnal variations in vertical distribution of a) UVPM$_{sec}$/UVPM and b) UVPM$_{sec}$/eBC ratios during the field campaign.

References

Han, S., Jang, M. Modeling daytime and nighttime secondary organic aerosol formation via multiphase reactions of biogenic hydrocarbons. Atmos. Chem. Phys., 23(2), 1209-1226, http://doi.org/10.5194/acp-23-1209-2023, 2023.

Kuang, Y., Luo, B., Huang, S., Liu, J. W., Hu, W. W., Peng, Y. W., Chen, D. H., Yue, D. L., Xu, W. Y., Yuan, B., Shao, M. Formation of highly absorptive secondary brown carbon through nighttime multiphase chemistry of biomass burning emissions. Atmos. Chem. Phys., 25(6), 3737-3752, http://doi.org/10.5194/acp-25-3737-2025, 2025.

Kulmala, M., Cai, R., Stolzenburg, D., Zhou, Y., Dada, L., Guo, Y., Yan, C., Petäjä, T., Jiang, J., Kerminen, V. M: The contribution of new particle formation and subsequent growth to haze formation, Environ. Sci.: Atmos., 2: 352–361, https://doi.org/10.1039/D1EA00096A, 2022.

Morgan, W. T., Allan, J. D., Bower, K. N., Capes, G., Crosier, J., Williams, P. I., Coe, H. Vertical distribution of sub-micron aerosol chemical composition from North-Western Europe and the North-East Atlantic. Atmos. Chem. Phys., 9, 5389–5401, https://doi.org/10.5194/acp-9-5389-2009, 2009.

Wang, H. C., Wang, H. L., Lu, X., Lu, K. D., Zhang, L., Tham, Y. J., Shi, Z. B., Aikin, K., Fan, S. J., Brown, S. S., Zhang, Y. H. Increased night-time oxidation over China despite widespread decrease across the globe. Nat. Geosci., 16: 217-223, https://doi.org/10.1038/s41561-022-01122-x, 2023.

Wu, C. and Yu, J. Z.: Determination of primary combustion source organic carbon-to-elemental carbon (OC / EC) ratio using ambient OC and EC measurements: secondary OC-EC correlation minimization method. Atmos. Chem. Phys., 16, 5453-5465, doi:10.5194/acp-16-5453-2016, 2016.

Zhao, S. P., Yu, Y., He, J. J., Dong, L. X., Qi, S. F. A New Physical Mechanism of Rainfall Facilitation to New Particle Formation. Geophys. Res. Lett., 51(1), e2023GL106842, https://doi.org/10.1029/2023GL106842, 2023b.

9. L330-L331, "To better … distinct features." Could you explain how you classified those profiles? Just based on your observation, or used some statistical model?

**Response:** Your question is highly pertinent, and implementing the necessary revisions

based on your query significantly enhances the clarity and readability of our manuscript. In this work, Since the observation heights at different times were not consistent, statistical models were not used. Instead, the vertical rate of change (or slope) of UVPM and eBC concentrations was analyzed. These profiles were grouped into four distinct categories: 'Uniformly decreasing (Cluster 1)', 'Non-uniformly decreasing (Cluster 2)', 'Double-inflection point (Cluster 3)', and 'Continuous increasing (Cluster 4)'. The detailed methodology for this specific clustering approach was, regrettably, omitted in the previous version of the manuscript. We have now integrated this comprehensive explanation into the revised version, which can be found in the updated manuscript. The specific modifications are as follows (Due to adjustments made to the figures within the manuscript, this illustration has been renumbered as "Figure 4"):

To better characterize vertical structure of UVPM and eBC within the stable PBL, the observed profiles were classified into four types, each exhibiting distinct features (Details of the method can be found in Text 2 of the Supplementary Material). Regarding the profile clustering presented in Figure 4, the classification was performed by analyzing the vertical change rates (or slopes) of the UVPM and eBC concentrations. These profiles were grouped into four distinct categories: Uniformly decreasing (Cluster 1), Non-uniformly decreasing (Cluster 2), Double-inflection point (Cluster 3), and Continuous increasing (Cluster 4).

**Text 2: Profile classification method**

In classifying the vertical profiles, we performed the cluster analysis based on the gradient variation of eBC (equivalent Black Carbon). The four resulting profile types were defined. Uniformly decreasing (Cluster 1): In the observed samples, the vertical gradient of eBC concentration is less than $0.6\ \mu g\ m^{-3}\ km^{-1}$ and remains nearly constant, exhibiting either a quasi-linear or weakly decreasing trend with height; Non-uniformly decreasing (Cluster 2): In the observed samples, the vertical gradient of eBC concentration exceeds $1.0\ \mu g\ m^{-3}\ km^{-1}$ within the lowest 500 m and persists over a vertical extent of at least 50 m, indicating a more rapid decrease with height; Double-inflection point (Cluster 3): Observation samples where the eBC concentration profile

clearly displayed two inflection points, typically characterized by an initial decrease with height, followed by an increase, and then a subsequent decrease; Continuous increasing (Cluster 4): Observation samples where the eBC concentration exhibited a continuous increase with increasing height across the observed column.

10. Figure 3. Is the black line represent the potential temperature? If yes, please match the color of the line and that of the upper x-axis text, or define it in your legend.

**Response:** Thank you for your suggestion. We have corrected the figure at the corresponding location in the manuscript. Due to adjustments made to the figures within the manuscript, this illustration has been renumbered as "Figure 4". The revised figure is shown below, where the blue and red lines represent the UVPM and eBC profiles, respectively, and the green line represents the potential temperature profile. Simultaneously, the upper x-axis representing potential temperature has also been changed to green for clarity.

[Figure]

Figure 4. Cluster analysis of UVPM and eBC concentration profiles during the field campaign. The blue, red, and green lines represent the profiles of UVPM, eBC, and potential temperature, respectively, and the shaded areas in the corresponding colors denote their standard deviations.

11. L360-362, "The AAE … Figure S7." Your discussion of AAE is unclear to me. How did you separate BC from other light-absorbing aerosols? Or is your AAE indeed the AAE for all aerosols? Please clarify that.

**Response:** Thank you for pointing out this issue. We have identified a writing error here. The AAE referred to in our Figure S8 actually represents the AAE value of the carbonaceous aerosols. We have corrected this statement in the manuscript.

The AAE, which characterizes the wavelength dependence of the mass-specific

absorption by carbonaceous aerosols (calculation details are given in Figure S7), was calculated for the study period; the average diurnal AAE values are shown in Figure S8.

12. L364-366, "We further … (Figure 4)." It is not clear to me how you defined events based on AAE. Is that based on the literature results? If so, please provide references. Otherwise, please explain how you did that.

**Response:** The pollution episodes analyzed in this study were identified based on on-site observational records, and the comparison of AAE values for carbonaceous aerosols under different pollution conditions was conducted using these observational records rather than relying on AAE values or related literature. To further support this identification approach, we have added detailed information on the environmental conditions in Figure S1, as described below. In addition, due to adjustments made to the figures in the manuscript, the corresponding figure has been renumbered as Figure 5.

We further selected observations from representative pollution events to compare AAE under different environmental conditions (Figure 5, Detailed explanations regarding the selection of pollution events are provided in Text 3 of the supplementary material).

**Text 3: Selection guidelines for pollution incidents**

1. Diesel vehicle emissions: During the 20:00 observation on July 17, 2024, a diesel harvester was operating approximately 50 m from our observation site and ceased work after about half an hour. Consequently, we define this specific observation as being affected by diesel vehicle emissions. For comparison, we selected the 23:00 observation—which has the shortest temporal difference—as the corresponding non-diesel vehicle emission observation.

2. Dust Pollution: A dust event occurred throughout the entire day of July 21, 2024. We defined the 8:00 observation, when the dust was most severe (PM$_{10}$ concentration in Pingliang urban area was 350 μg m$^{-3}$ at that time), as the dust pollution observation. Since the dust event lasted for a long duration, we chose the observation taken at the

same time on the previous day (8:00 on July 20, 2024) for comparison.

[Figure]

Figure S1. Temporal variations of meteorological parameters, including air temperature (T), relative humidity (RH), wind direction (WD), wind speed (WS), and air pollutants, including particulate matter with aerodynamic diameter smaller than 2.5 μm ($PM_{2.5}$) and than 10 μm ($PM_{10}$), nitrogen dioxide ($NO_2$), ozone ($O_3$), sulfur dioxide ($SO_2$), and carbon monoxide (CO), were observed during the monitoring period. The shaded areas in the figure represent the during and after vehicle emissions, no dust pollution, and dust pollution at 20:00 on July 17, 23:00 on July 17, 2024, 08:00 on July 20, 2024, 08:00 on July 21, 2024, respectively.

[Figure]

Figure 5. AAE profiles under different pollution conditions (Diesel vehicle emissions, after diesel vehicle emissions, no dust pollution, and dust pollution occurred at 20:00 on July 17, 2024; 23:00 on July 17, 2024; 08:00 on July 20, 2024; 08:00 on July 21, 2024, respectively).

13. L367-368, "Likewise, … diesel contributions." Which one is a diesel emission event?

The vehicle emission line? If so, did you include gasoline vehicle emissions? How did you identify that event?

**Response:** Thank you for your question. Our reply regarding this issue is as follows:

During the 20:00 observation on July 17, 2024, a diesel harvester was operating approximately 50 m from our observation site and ceased work after about half an hour. Consequently, we define this specific observation as being affected by Diesel vehicle emissions. For comparison, we selected the 23:00 observation—which has the shortest temporal difference—as the corresponding non-motor vehicle emission observation.

14. L368-370, "Under heavy … vapor aloft." I do not trust your measurements under heavy fog since high RH biases MA350 measurements.

**Response:** Thank you for your comment. In Fig. S1, we plotted the humidity recorded by the MA350 for the samples. It can be seen that during the observation period, the sample humidity did not exceed 60%, and we consider the measurements to be reliable under these conditions.

[Figure]

Figure S1. Temporal variations of meteorological parameters, including air temperature (T), relative humidity (RH), wind direction (WD), wind speed (WS), and air pollutants, including particulate matter with aerodynamic diameter smaller than 2.5 μm ($PM_{2.5}$) and than 10 μm ($PM_{10}$), nitrogen dioxide ($NO_2$), ozone ($O_3$), sulfur dioxide ($SO_2$), and carbon monoxide (CO),

were observed during the monitoring period. The shaded areas in the figure represent during and after vehicle emissions, no dust pollution, and dust pollution at 20:00 on July 17, 23:00 on July 17, 2024, 08:00 on July 20, 2024, 08:00 on July 21, 2024, respectively.

15. 370-373, "Hence, … profile." I am not quite sure where this comes from. It seems this sentence was not connected to the previous discussions.

**Response:** Thank you for your suggestion. We agree that this section's discussion has poor relevance to the preceding topic. To ensure the logical coherence of the study, we have removed this section of the discussion.

16. L375-377, "Previous … (Figure S8)". Please add references and the calculations you did.

**Response:** Thank you for your suggestion. We have added the method description to the Supporting Information. The specific content is as follows:

Previous studies commonly employ a two-component model to differentiate the Absorption Ångström Exponent of equivalent black carbon ($AAE_{eBC}$) and brown carbon ($AAE_{UVPM}$), fixing $AAE_{eBC}$ at 1 and thereby deriving $AAE_{BrC}$ (Figure S9, The calculation methodology is provided in Text 4 of the supplementary material) (Chen et al., 2015; Chow et al., 2018). Elevated values of $AAE_{UVPM}$ are typically indicative of biomass burning and secondary aging processes (Olson et al., 2015; Gombi et al., 2025).

**Text 4: Two-component model**

Referring to the studies by Chen et al. (2015), we used a two-component model to separate the light absorption coefficients of eBC and UVPM, calculated using the following formulas:

$$b_{Abs, \lambda} = b_{Abs, \lambda, eBC} + b_{Abs, \lambda, UVPM} = A_1 \times \lambda^{-AAE_{eBC}} + A_2 \times \lambda^{-AAE_{UVPM}} \tag{3}.$$

Where $AAE_{eBC}$ and $AAE_{UVPM}$ are the Absorption Ångström Exponents (AAE) for eBC and UVPM, respectively. In the calculation process, eBC is typically assumed to be composed of pure carbon, thus $AAE_{eBC}$ is approximated as 1.0 (Chow et al., 2018). The

values of $A_1$, $A_2$, and $AAE_{UVPM}$ in the equations are then determined through a least-squares fitting method.

References

Chen, L.W.A., Chow, J.C., Wang, X.L., Robles, J.A., Sumlin, B.J., Lowenthal, D.H., Zimmermann, R., Watson, J.G. Multi-wavelength optical measurement to enhance thermal/optical analysis for carbonaceous aerosol. Atmos. Meas. Tech., 8(1), 451-461, https://doi.org/10.5194/amt-8-451-2015, 2015.

Chow, J.C., Watson, J.G., Green, M.C., Wang, X.L., Chen, L.-W.A., Trimble, D.L., Cropper, P.M., Kohl, S.D., Gronstal, S.B. Separation of brown carbon from black carbon for IMPROVE and CSN PM2.5 samples, Air Waste Manag. Assoc., 68(5), 494-510, https://doi.org/10.1080/10962247.2018.1426653, 2018.

17. L377-381, "An $AAE_{BrC}$ … observation site." Please add references.

**Response:** Thank you for your suggestion. We have revised this statement and cited the relevant references. The revised content is as follows:

Elevated values of $AAE_{UVPM}$ are typically indicative of biomass burning and secondary aging processes (Olson et al., 2015; Gombi et al., 2025).

References

Olson, M. R., Victiria, G. M., Robinson, M. A., Rooy, P. V., Dietenberger, M. A., Bergin, M., Schauer, J. J. Investigation of black and brown carbon multiple-wavelength-dependent light absorption from biomass and fossil fuel combustion source emissions. J. Geophys. Res.-Atmos., 120(13): 6682-6697. https://doi.org/10.1002/2014JD022970, 2015.

Gombi, C., Rahman, A., Hodovány, S. et al. A demonstrative study of a novel approach for spectral based source apportionment of ambient aerosols. Sci. Rep., 15, 19501. https://doi.org/10.1038/s41598-025-04022-3, 2025.

18. L394-396, "From the … Shi et al., 2020)." How did you derive PBLHc? How did you get water vapor and aerosols? Where are your PBLHc results?

**Response:** Thank you very much for your question/inquiry. As the calculation of $PBLH_C$ was not involved in this study, we have deleted the corresponding content. The revised content is as follows:

Diurnal evolution of the PBLH is one of the primary factors controlling aerosol vertical distribution and is essential for understanding feedback between the PBL meteorology and aerosols. Because PBLH evolves continuously, we selected a day with uninterrupted observations; however, owing to strong upper-level winds, continuous carbonaceous aerosol profiles were obtained only on 27 July 2024 at seven time slots (Figure 6). Hence, this day serves as a case study for examining how diurnal PBL evolution influences aerosol vertical structure. The parcel method is well suited to convective conditions but cannot accurately resolve PBLH during early morning and nighttime (Holzworth, 1964), whereas it performs reliably under strong daytime convection. Using this approach, PBLH at 11:00 and 17:00 LT were 503 m and 459 m, respectively; at 14:00, measurement ceilings did not reach PBLH. By contrast, the potential temperature gradient method yields accurate estimates at night and in the early morning: PBLH at 05:00, 08:00, 20:00, and 23:00 LT were 255 m, 181 m, 260 m, and 216 m, respectively.

Analysis of the potential temperature profiles in Figure 6 indicates that at around 05:00 LT the PBL top lay near 260 m. Below this altitude, both eBC and UVPM concentrations decrease slightly with height. The potential temperature profile further reveals a deep residual layer above the PBL top, where colder near-surface air is trapped beneath warmer air aloft, creating a stable stratification that inhibits mixing within that layer and leads to increasing particle concentrations toward its base. In the transition to the free troposphere above the residual layer, comparatively low aerosol concentrations and enhanced turbulence promote further dilution, and beyond approximately 500 m eBC and UVPM again decline with height. By 08:00 LT the PBL remained near 200 m, and the vertical variation in aerosol concentrations mirrored the pattern observed at 05:00. With increasing solar insolation, however, surface heating intensified convection so that by 11:00 LT the PBL top had risen to roughly 500 m. During this stage, relatively small vertical gradients in eBC and UVPM within the PBL indicate well-mixed conditions. At 14:00 LT tethered-balloon sampling did not reach the PBL top, but observations within the PBL show uniform aerosol distributions, preventing a direct

assessment of PBLH effects on concentration profiles. The potential temperature gradients between 11:00 and 14:00 LT exhibit significant fluctuations, signaling unstable stratification favorable to vertical pollutant transport (Li, 2019). From 14:00 to 17:00 LT, as solar radiation waned and surface temperatures fell, the PBL top subsided to about 450 m. At this time, potential temperatures within the PBL remained lower than at the surface and displayed pronounced variability, reflecting continued unstable stratification and strong vertical mixing; eBC and UVPM maintained nearly uniform distributions. Above the PBL, a mixed layer approximately 300 m thick persisted; at its top, diminished turbulence inhibited aerosol dispersion, causing localized accumulation of eBC and UVPM (Ding et al., 2016). By 20:00 LT, sunset-driven surface cooling weakened convection, a nocturnal temperature inversion developed near the ground, and calm winds led to pollutant accumulation at low altitudes. As surface temperatures continued to drop, vertical transport further diminished, confining aerosols below roughly 200 m by 23:00 LT.

[Figure]

Figure 6. Diurnal variations in profiles of eBC, UVPM and potential temperature on July 27, 2024. The red line represents eBC, the blue line represents UVPM, and the green line represents the potential temperature profile. The light orange shaded area represents the difference between UVPM and eBC. The black dashed lines represent the PBLH.

19. L432-434, "At 14:00 … boundary-layer-height." It is unclear to me since your PBLH is below 200m.

**Response:** Thank you for your question. Following the descriptions in Holzworth (1964) and Zhang et al. (2020), the planetary boundary layer height during daytime in this study is determined using the parcel method, whereas the value shown at 14:00 in Figure 6 was derived using the potential temperature gradient method. Since different methods were applied to daytime and nighttime periods, the original Figure 6 did not clearly distinguish between them, so we have revised the figure accordingly. For 05:00, 08:00, 20:00, and 23:00, only the gradient-method results are retained; for 11:00 and 17:00, only the parcel-method results are shown. At 14:00, the observed potential temperature profile did not extend high enough to calculate the parcel-method PBLH, and therefore no PBLH is available for that time. The revised figure is presented as follows:

[Figure]

Figure 6. Diurnal variations in profiles of eBC, UVPM and potential temperature on July 27, 2024. The red line represents eBC, the blue line represents UVPM, and the green line represents the potential temperature profile. The light orange shaded area represents the difference between UVPM and eBC. The black dashed lines represent the PBLH.

References

Holzworth, G. C. Estimates of mean maximum mixing depths in the contiguous united states. Mon. Weather Rev., 92(5), 235-242, http://doi.org/10.1175/ 1520-

0493(1964)092<0235:EOMMMD>2.3.CO;2, 1964.

Zhang, H. S., Zhang, X. Y., Li, Q. H., Cai, X. H., Fan, S. J., Song, Y., Hu, F., Che, H. Z., Quan, J. N., Kang, L., Zhu, T. Research progress on estimation of atmospheric boundary layer height. Acta Meteorol. Sin., 78(3), 522-536, http://doi.org/10.11676/qxxb2020.044, 2020.

20. Figure 5, I don't see PBLH (Parcel) dash lines in 5:00, 8:00, 20:00, and 23:00. Please explain.

**Response:** Thank you for your question. Due to adjustments made to the figures within the manuscript, this illustration has been renumbered as "Figure 6". The definition of the Parcel Method (which is based on the Maximum Mixing Depth, MMD) given by Holzworth (1964) is: "The height at which a dry adiabatic line extending upward from the maximum surface temperature intersects the most recently observed temperature profile." Therefore, this method is appropriate for unstable atmospheric conditions and yields more accurate results for calculating the PBLH during the daytime. Consequently, when using this method, the PBLH could not be obtained for the 05:00, 08:00, 20:00, and 23:00 observation periods; the PBLH for these time slots was instead calculated using the potential temperature gradient method. The applicable conditions for both methods are discussed in the study by Zhang et al. (2020), which has been cited in this manuscript. This explanation has been added to Section 2.3.

Planetary boundary layer is the part of the atmosphere closest to the planet's surface, accounting for approximately 10% – 20% of the troposphere. It is the lowest layer of the troposphere directly influenced by surface forcing, with a response time of less than one hour, playing a critical role in the dispersion and transport of air pollutants. Within the PBL, turbulent mixing processes homogenize air temperature and humidity, resulting in relatively uniform distributions of these properties. PBL height (PBLH) refers to the thickness of the layer most significantly affected by the surface. In other words, it is the vertical extent that surface turbulent motion (caused by surface heating, friction, or topography, etc.) can effectively influence (Emeis et al., 2008; Seibert et al., 2000). This study employs two established methodologies for determining the PBLH:

the potential temperature gradient method and the parcel method (Holzworth, 1964; Zhang et al., 2020). The potential temperature gradient method was utilized for calculating the PBLH during the nighttime and early morning hours (20:00, 23:00, 05:00, and 08:00 LT), while the parcel method was applied specifically for the daytime periods (11:00, 14:00, and 17:00 LT). Detailed computational procedures for both approaches are summarized in Supplementary Table S1.

References

Holzworth, G. C. Estimates of mean maximum mixing depths in the contiguous united states. Mon. Weather Rev., 92(5), 235-242, http://doi.org/10.1175/ 1520-0493(1964)092<0235:EOMMMD>2.3.CO;2, 1964.

Zhang, H. S., Zhang, X. Y., Li, Q. H., Cai, X. H., Fan, S. J., Song, Y., Hu, F., Che, H. Z., Quan, J. N., Kang, L., Zhu, T. Research progress on estimation of atmospheric boundary layer height. Acta Meteorol. Sin., 78(3), 522-536, http://doi.org/10.11676/qxxb2020.044, 2020.

21. L462-465, "It shows that … receptor site." Please explain this in more detail, as the air masses in general originated from high altitudes and did not exhibit any interaction with lower emissions in the surrounding urban areas.

**Response:** Thank you for your comments. In Fig. S10, we plotted all air mass trajectories arriving at the observation site at 500 m during the study period, with different colors along each trajectory indicating the altitude of the air parcel. In addition, we included the planetary boundary layer height (PBLH) of the nearest urban area of Pingliang during the observation period (Fig. S11, PBLH obtained from the Global Data Assimilation System). As shown in the figures, when the air masses reached the vicinity of the observation site, their altitudes consistently decreased to below 1500 m, while the PBLH of the surrounding urban area was above 1500 m for most of the time. This indicates that the air masses may have carried pollutants from nearby urban emissions to the observation site. The revised content is as follows:

Long-range transport of air masses plays a crucial role in shaping the vertical distribution of air pollutants. Figure S10 presents the 500 m backward trajectories and

their altitude profiles for air masses arriving at the site during the observation period, and Figure S11 presents temporal variation in the PBLH of the nearest urban area of Pingliang during the observation period. Figure S11 shows that, upon entering the Pingliang region, these air parcels generally descend to below PBLH, meaning that, in addition to pollutants carried within the air mass itself, emissions from surrounding urban areas also significantly impact the receptor site.

[Figure]

Figure S10. All 48-hour backward trajectories of air masses that reached the Pingliang Observatory at 500 m height (above ground level) during the measurement period (from July 15 to 24, 2023, and from July 17 to 31, 2024), where the different colors of each trajectory represent the altitude of the air mass at different geographic locations.

[Figure]

Figure S11. Temporal variation of the planetary boundary layer height over Pingliang urban area during the observation period (PBLH obtained from the Global Data Assimilation System). The observations in the year of 2023 and 2024 were separated by the vertical blue dashed line.

22. 5-468, "Trajectory-cluster … " Fig. S9 shows numerous air masses originating from Shanxi and Sichuan. Why don't you have those in your sources analysis? What are the

color bar and color lines? How did you do the cluster analysis?

**Response:** Thank you for reviewing our study and for raising this question.

(1) During the backward trajectory clustering analysis, we employed the "Cluster Analysis" module in the MeteoInfo software. The official documentation is available at the following link: https://www.meteothink.org/docs/trajstat/cluster_cal.html. For clustering the trajectory profiles, we used the Euclidean distance method (Sirois and Bottenheim, 1995), and selected four clusters to identify the dominant source regions of the air masses arriving at the observation site. The Euclidean distance between two backward trajectories is defined as follows:

$$d_{12} = \sqrt{\sum_{i=1}^{n} ((X_1(i)\text{-}X_2(i))^2 + ((Y_1(i)\text{-}Y_2(i))^2},$$

where $X_1$ ($Y_1$) and $X_2$ ($Y_2$) refer to backward trajectories 1 and 2, respectively.

(2) The apparent loss of some air masses originating from the Shanxi and Sichuan regions after clustering is primarily because these air masses constitute a low proportion among the total air masses from those directions. Consequently, their characteristic information was masked after the clustering process. It is important to note that these air masses were included in the WPSCF and WCWT analyses. Therefore, the result presented here is considered reasonable.

(3) In addition, the different colors in the figure represent the altitude (above ground level) of the trajectory at different geographic locations during its movement. This information has been described in the figure caption to enhance readability.

[Figure]

Figure S10. All 48-hour backward trajectories of air masses that reached the Pingliang

Observatory at 500 m height (above ground level) during the measurement period (from July 15 to 24, 2023, and from July 17 to 31, 2024), where the different colors of each trajectory represent the altitude of the air mass at different geographic locations.

Reference

Sirois, A., Bottenheim, J. W. Use of backward trajectories to interpret the 5-year record of PAN and O3 ambient air concentrations at Kejimkujik National Park, Nova Scotia. J. Geophys. Res.-Atmos., 100(D2), 2867-2881, https://doi.org/10.1029/94JD02951, 1995.

23. Figure 6. The color bar title is confusing since it is WPSCF, neither PSCF nor CWT.

**Response:** Thank you for pointing out this issue. The WPSCF (Weighted Potential Source Contribution Function) and WCWT (Weighted Concentration Weighted Trajectory) used in this study are calculated by applying weighting to the PSCF and CWT, respectively. This treatment can effectively reduce the errors caused by a small number of trajectories. The relevant calculations have been added in the Text 1 of supplementary material and are presented as follows (Due to adjustments made to the figures within the manuscript, this illustration has been renumbered as "Figure 7"):

The shaded overlays in Figure 7 show the weighted potential source contribution function (WPSCF) and weighted concentration-weighted trajectory (WCWT) results for UVPM. WPSCF indicates that air parcels from Inner Mongolia, Ningxia, Shanxi, and Shaanxi to the north, east, and south exert the greatest influence on the Pingliang site. Specifically, at 100 m the highest WPSCF values occur along the Shaanxi–Gansu border, while at 500 m parcels from central Shaanxi and the Shanxi–Henan–Shaanxi nexus dominate, followed by the tri-provincial junction of Shaanxi, Gansu, and Sichuan. However, the WCWT analysis identifies Hanzhong in southern Shaanxi as the most significant source region for UVPM at the receptor. Taken together, WPSCF and WCWT pinpoint southern Shaanxi cities and local emissions around Pingliang as major contributors to carbonaceous aerosol pollution, whereas at higher altitudes UVPM is primarily transported from the south.

**Text 1: Analysis methods of WPSCF and WCWT**
In the calculation of WPSCF (weighted potential source contribution function) and

WCWT (weighted concentration weighted trajectory), we first calculated the air mass trajectories for the preceding 48 hours at heights of 100 m, 500 m, and 1000 m within all observation days (July 15–24, 2023, and July 17–31, 2024). Next, the UVPM mass concentration data were overlaid onto the air mass trajectory data, and the areas traversed by the air mass trajectories were divided into a 0.5°×0.5° grid, in order to determine the potential source regions of UVPM for the observation station. The calculation methods for WPSCF and WCWT are shown in Equations (1) and (2) (Wang, 2024):

$$WPSCF_{ij} = \frac{m_{ij}}{n_{ij}} \times W_{ij} \tag{1},$$

$$C_{ij} = \frac{1}{\sum_{l=1}^{M} \tau_{ijl}} \times \sum_{l=1}^{M} C_l \tau_{ijl} \tag{2}.$$

In Equation (1), $m_{ij}$ represents the number of polluted trajectory endpoints within the ij-th cell, and $n_{ij}$ represents the total number of trajectory endpoints within the ij-th cell. $W_{ij}$ is the weighting factor. In Equation (2), $C_{ij}$ is the average weighted concentration in the ij-th cell, l is the index of the trajectory, M is the total number of trajectories, $C_l$ is the concentration observed on arrival of trajectory l, and $\tau_{ijl}$ is the time spent in the ij-th cell by trajectory l. A high value for $C_{ij}$ implies that air parcels traveling over the ij-th cell would be, on average, associated with high concentrations at the receptor. The arbitrary weighting function described above was also used in the CWT analyses to reduce the effect of the small values of $n_{ij}$. (All the above content refers to the official MeteoInfo documentation, which can be found at the following URL: http://www.meteothink.org/docs/trajstat/trajstatrun.html.)

24. L482-484, "Overall, observation site." Your CPF results come out in the next section. You haven't discussed that yet, so how can you make the comparison?

**Response:** Thank you very much for raising this question. In the revised manuscript, we have moved this analysis to follow the CPF results for UVPM and eBC.

25. Figure 7. What is the color bar? Did you plot a similar figure for eBC?

**Response:** Thank you for your valuable suggestion. Due to adjustments made to the figures within the manuscript, this illustration has been renumbered as "Figure 8". We have added additional information regarding the CPF analysis of eBC in Section 3.3.2 and in Figure 8 of the revised manuscript. We have also redrawn the CPF plot at this location. revised content is presented below:

3.3.2 Impacts of local sources on carbonaceous aerosols

In addition to long-range transport, local wind speed and direction significantly modulate the vertical distribution of aerosols at the observation site. Conditional Probability Function (CPF) plots, which relate pollutant concentrations to wind sectors and speeds, help elucidate these local transport and dispersion mechanisms. Figure 8 shows CPF diagrams for the 90[th] percentile of UVPM and eBC concentrations at 100 m, 500 m and 1000 m. The mean UVPM concentrations decrease with altitude, from 0.86 μg m$^{-3}$ at 100 m, to 0.63 μg m$^{-3}$ at 500 m, and to 0.44 μg m$^{-3}$ at 1000 m, indicating that extreme UVPM events become less probable at higher altitudes. At 100 m, elevated UVPM concentrations are associated with southerly, southwesterly, southeasterly, northerly, and northwesterly winds, reflecting the combined influence of urban emissions and local rural emissions surrounding the observation site. At 500 m, the influence of northwesterly, southwesterly and northerly sectors diminishes, while southerly winds remain the dominant driver of high UVPM, albeit with reduced effect. Southeasterly winds occasionally lead to relatively high UVPM values at both 100 m and 500 m altitude, although this occurrence is less frequent. We hypothesize that this variability may be associated with local and sporadic emissions from residents in the vicinity, as the observation site is located in a rural area with no stationary point sources. However, the specific cause cannot be definitively determined by the CPF alone. We will continue to analyze this phenomenon in subsequent observational studies to provide a clearer explanation.

The analysis of eBC shows results similar to those of UVPM. The main difference is that at 500 m and 1000 m, southeasterly winds exert a slightly stronger influence on UVPM than on eBC. In addition, the relationships between UVPM (or eBC) and wind

direction and speed at different altitudes indicate that carbonaceous aerosols at 100 m above the observation site are highly sensitive to almost all wind directions. This is primarily because pollutants at this height are strongly affected by local emissions from rural areas surrounding the site. In contrast, at 500 m and 1000 m, carbonaceous aerosol concentrations exhibit a pronounced association with southerly and southeasterly winds, while their dependence on other wind directions weakens. This indicates that transport from the south and southeast is the significant source of the carbonaceous aerosol burden above the observation site, a pattern that is also reflected in the WPSCF and WCWT results shown in Figure 7. In addition, this may also be related to the fact that Pingliang urban area is located south of the observation site (Figure 1).

[Figure]

Figure 8. CPF (Conditional Probability Function) diagrams of the 90th percentile of the UVPM and eBC concentrations with respect to wind speed and direction at the heights of 100 m, 500 m and 1000 m above ground level.

26. Figure 8. Please change the x-axis ticks to time only. You don't need to show year, month, and day. How did you calculate each arrow? Averaged by how many altitudes or time? Please also label the boundary layer height. Your boundary layer was low that day. What are the lowest altitudes your lidar can measure confidently?

**Response:** (1) Due to adjustments made to the figures within the manuscript, this

illustration has been renumbered as "Figure 9". We agree with the modifications you proposed and have revised Figure 9 accordingly. In Figure 9, the x-axis has been modified to display time only, and the PBLH is indicated using a black dashed line. Each arrow represents the mean wind vector calculated for a given height (or time) interval.

(2) Specifically, for each individual observation, the wind vector was first decomposed into its zonal (u) and meridional (v) components. The u and v components were then averaged separately over all valid samples within the corresponding interval. Finally, the averaged u and v components were recombined to obtain the mean wind vector, which is displayed as an arrow with its direction indicating the mean wind direction and its length proportional to the mean wind speed.

(3) Furthermore, a description of the wind direction represented by the arrows has been added to the legend: The direction of the arrow represents the wind vector direction, i.e., a rightward arrow represents a westerly wind. In addition, we have included information regarding the vertical resolution of the wind speed and direction (the resolution is 50 m, with a blind zone in the bottom 100 m) and the time used for the $V_{\text{TKE}}$ calculation (the lidar is set up to scan wind speed at different altitudes from 00 minutes to 05 minutes of every hour, thus our $V_{\text{TKE}}$ is calculated using wind speed data collected within this 5-minute interval). During each observation period on July 27, 2024, the lidar observation height consistently reached 2000 m. The revised figure and caption are as follows:

[Figure]

Figure 9. a) Vertical wind speed, horizontal wind speed, and wind direction at heights of 100 – 1500 m during different periods on July 27, 2024 (The direction of the arrow represents the wind vector direction, i.e., a rightward arrow represents a westerly wind). The height resolution for wind direction and speed is 50 m, and the length of the arrow represents the magnitude of the horizontal wind speed. The color of the arrows represents the vertical wind speed. b) $V_{TKE}$ values at different heights (calculated based on wind speed data within 5 min), where different colors represent different $V_{TKE}$ values. The black dashed line in the figure represents the PBLH during the observation period.

---

## Author Comment (AC2)

**Response to Comments from Reviewer**

**Journal:** Atmospheric Chemistry and Physics

**Manuscript ID:** egusphere-2025-3254

**Title:** " Measurement report: Measurement report: Impacts of Thermodynamic and Dynamic Processes on the Vertical Distribution of Carbonaceous Aerosols: lessons from in-situ observations at eastern foothills of LiuPan Mountains, Loess Plateau"

**Author(s):** Shaofeng Qi, Suping Zhao*, Ye Yu, Longxiang Dong, Tong Zhang, Guo Zhao, Jianglin Li, Xiang Zhang, and Yiting Lv

*Correspondence to*: Suping Zhao (zhaosp@lzb.ac.cn)

In the response, the content in **black** font represents the original text from the manuscript, the content in **blue** font indicates additions or revisions, and the content in **red** font is the response to the reviewers' questions and suggestions.

**Dear Editors and Reviewers:**

Thank you for your and the reviewers' comments concerning our manuscript entitled "Impacts of Thermodynamic and Dynamic Processes on the Vertical Distribution of Carbonaceous Aerosols: lessons from in-situ observations at eastern foothills of LiuPan Mountains, Loess Plateau" (https://doi.org/10.5194/egusphere-2025-3254). Those comments are all very valuable, greatly assisting us in revising and improving our paper, as well as providing important guidance for our research. We have carefully studied the comments and made corrections accordingly, which can be viewed in the revised version of our manuscript. The main corrections in the paper and the responses to the reviewer's comments are provided below:

**Major Concerns:**

1. The definitions of the terms "UVPM", "BC", and "IRBC" in this study are not clear and thus might influence the interpretation of data and cause confusion for the readers. The terms are also used inconsistently throughout the manuscript, raising concerns of logical conflict in the study. In Line 20, UVPM seems to be the acronym for "ultraviolet particulate matter" while the other term "ultraviolet absorbing particulate matter" also

appears to be the full form of UVPM in Line 605. More details are described as separate bullet points listed below.

**Response:** We sincerely appreciate this constructive suggestion, which has substantially improved the clarity and consistency of our manuscript. In response, we have carefully revised the relevant sections. Specifically, the concentration measured at 880 nm is now uniformly defined as equivalent black carbon (eBC), and all previous instances of "BC" or "IRBC" have been corrected accordingly. In addition, to avoid any potential ambiguity, and following the official documentation of the MA350 instrument (URL: https://aethlabs.com/products/ma300, last accessed: 13 November 2025), we have standardized the terminology for the species measured at 375 nm by defining them as ultraviolet-absorbing particulate matter (UVPM).

The average near-surface concentrations of equivalent black carbon (eBC) and ultraviolet-absorbing particulate matter (UVPM) in Pingliang were 0.84 μg m$^{-3}$ and 1.24 μg m$^{-3}$, respectively.

2. Lines 159 – 161: Is black carbon (BC) concentration measured at wavelength 880 nm denoted as BC or IRBC? From the context of the whole manuscript, it seems that BC represents the particle concentration measured at wavelengths 470 nm, 528 nm, 625 nm, and 880 nm, while IRBC represents only the particle concentration measured at wavelength 880 nm, and the particle concentration measured at wavelength 375 is denoted as UVPM. Please ensure consistency throughout the manuscript and avoid confusion for the readers.

**Response:** Thank you for pointing out the deficiencies in this section. We have implemented detailed revisions regarding this issue in the manuscript: the concentration observed at 880 nm (IR) is now defined as equivalent BC (denoted as "eBC"), while measurements at 375 nm (UV) represent ultraviolet-absorbing particulate matter (UVPM). We have corrected all instances of "BC", "IRBC" and "UVPM" in the text.

3. Lines 161 – 163: Where does this definition of UVPM come from? Please provide relevant citations or explicitly mention that it is specifically defined by this study as the

term "ultraviolet particulate matter" and "ultraviolet absorbing particulate matter" are not yet a standardized term in the literature. Please also provide the reason for using this term to convince the readers if using this term is necessary to be created as opposed to using the more commonly used brown carbon or dust aerosol when it comes to absorption. Based on the context of the whole manuscript, the term UVPM seems to be the total absorption contributed from dust, brown carbon, and black carbon at the wavelength 375 nm, but it would be great if this could be clarified by the authors.

**Response:** Thank you for your suggestion. The observation period corresponds to the local summer season. Except for one special dust event, the air quality in this region was generally good, with no evident dust pollution. According to the instrument's official specifications and existing studies (Zhao et al., 2023a), the material detected at the 375 nm wavelength includes both primary organic carbon and black carbon directly emitted from sources such as biomass burning and coal combustion, as well as secondary organic carbon formed through various atmospheric chemical reactions. Therefore, in this study, we refer to this component as ultraviolet-absorbing particulate matter (UVPM). This information is derived from Section 1.2, paragraph 2 of the official manual for the MicroAeth® MA350 aethalometer, available at: https://aethlabs.com/products/ma300. The specific revisions are as follows:

The instrument employs five laser wavelengths: 375 nm (UV), 470 nm (Blue), 528 nm (Green), 625 nm (Red), and 880 nm (IR). Carbon concentrations measured at 880 nm are considered to represent the equivalent black carbon concentration (denoted as eBC) (Zhao et al., 2023a), whereas those measured at 375 nm correspond to ultraviolet-absorbing particulate matter (UVPM), which includes primary brown carbon and black carbon emitted from sources such as biomass burning and coal combustion, as well as secondary brown carbon formed through atmospheric oxidation processes (The relevant introduction/details regarding the substance measured at a wavelength of 375 nm can be referenced in the official manual of the MicroAeth® MA Series MA300 instrument. URL: https://aethlabs.com/products/ma300, last accessed: 13 November 2025).

Reference

Zhao, S. P., He, J. J., Dong, L. X., Qi, S. F., Yin, D. Y., Chen, J. B., Yu, Y. Contrasting Vertical Circulation between Severe and Light Air Pollution inside a Deep Basin Results from the Collaborative Experiment of 3D Boundary-Layer Meteorology and Pollution at the Sichuan Basin (BLMP-SCB). Bull. Am. Meteorol. Soc., 104(2), E411-E434, http://doi.org/10.1175/BAMS-D-22-0150.1, 2023a.

4. Lines 176 – 178: If only the absorption at wavelength 880 nm is considered as black carbon according to the statement in the previous section (Lines 159 – 161), why are the absorptions at multiple wavelengths (470 nm, 528 nm, and 625 nm) used for the calculation of light absorption coefficient of black carbon? Please revise the corresponding statements to avoid conflicts of logic throughout the manuscript.

**Response:** Thank you for pointing out this error. This was a typographical error on our part. The absorption coefficient calculated here should be the light absorption coefficient of carbonaceous aerosol at different wavelengths. We have corrected the content in the manuscript accordingly:

$$b_{Abs} = MAC(\lambda) \times [C] \tag{1},$$

where $MAC(\lambda)$ denotes the mass absorption cross-section of carbonaceous aerosols at specific wavelengths. For the MicroAeth® series instruments, the MAC values at 375 nm, 470 nm, 528 nm, 625 nm, and 880 nm are 24.07 $m^2\,g^{-1}$, 19.07 $m^2\,g^{-1}$, 17.03 $m^2\,g^{-1}$, 14.09 $m^2\,g^{-1}$, and 10.12 $m^2\,g^{-1}$, respectively (Zhao et al., 2023a). The [C] represents the concentration of carbonaceous aerosol at different wavelengths.

Additionally, the light absorption coefficient of BC can be expressed as:

$$b_{Abs}(\lambda) = k \times \lambda^{-AAE} \tag{2}.$$

In Eq. (2), k is a wavelength-independent constant, and AAE represents the Absorption Ångström Exponent, which characterizes the wavelength dependence of light absorption of carbonaceous aerosol (Ångström, 1929). A higher AAE value signifies that the aerosol absorption capacity decreases more rapidly with increasing wavelength.

By combining the $b_{Abs}$ values derived from Eq. (1) with Eq. (2), vertical profiles of AAE for carbonaceous aerosols were obtained.

Reference:

Ångström, A., 1929. On the atmospheric transmission of sun radiation and on dust in the air. Geogr. Ann. 11, 156–166.

**Minor Concerns:**

1. Lines 156 – 158: Does this instrument "MicroAeth® MA350" attribute all light absorption to black carbon? How do the authors quantitatively distinguish black carbon from other light-absorbing aerosols? What is the corresponding uncertainty for this method to quantify black carbon?

**Response:** Thank you for your question, which allows us to clarify our methodology. (1) In this study, the MicroAeth® MA350 aethalometer was utilized, which provides measurements of carbonaceous aerosols at five different wavelengths using an optical method. Specifically, the aerosol concentration detected at the wavelength of 880 nm is defined as equivalent black carbon (eBC), while the concentration at 375 nm is designated as ultraviolet-absorbing particulate matter (UVPM). The technical specifications of the MicroAeth® MA350 monitor are elaborated in Section 1.2 (second paragraph) of the official user manual, which is available at the following link: https://aethlabs.com/products/ma300. To summarize, the light absorption discussed in this study refers to that of carbonaceous aerosols, which consist of both eBC and brown carbon (BrC). (2) A two-component model was applied in this work to quantitatively distinguish the light absorption contributions from eBC and UVPM, respectively. Detailed methodological descriptions of the two-component model have been supplemented in Text 4 of the supplementary materials.

**Text 4: Two-component model**

Referring to the studies by Chen et al. (2015), we used a two-component model to separate the light absorption coefficients of eBC and UVPM, calculated using the following formulas:

$$b_{Abs, \lambda} = b_{Abs, \lambda, eBC} + b_{Abs, \lambda, UVPM} = A_1 \times \lambda^{-AAE_{eBC}} + A_2 \times \lambda^{-AAE_{UVPM}} \qquad (3).$$

Where $AAE_{eBC}$ and $AAE_{UVPM}$ are the Absorption Ångström Exponents (AAE) for eBC and UVPM, respectively. In the calculation process, eBC is typically assumed to be composed of pure carbon, thus $AAE_{eBC}$ is approximated as 1.0 (Chow et al., 2018). The values of $A_1$, $A_2$, and $AAE_{UVPM}$ in the equation are then determined through a least-squares fitting method.

References

Chen, L.W.A., Chow, J.C., Wang, X.L., Robles, J.A., Sumlin, B.J., Lowenthal, D.H., Zimmermann, R., Watson, J.G. Multi-wavelength optical measurement to enhance thermal/optical analysis for carbonaceous aerosol. Atmos. Meas. Tech., 8(1), 451-461, https://doi.org/10.5194/amt-8-451-2015, 2015.

Chow, J.C., Watson, J.G., Green, M.C., Wang, X.L., Chen, L.-W.A., Trimble, D.L., Cropper, P.M., Kohl, S.D., Gronstal, S.B. Separation of brown carbon from black carbon for IMPROVE and CSN PM2.5 samples, Air Waste Manag. Assoc., 68(5), 494-510, https://doi.org/10.1080/10962247.2018.1426653, 2018.

(3) Since the uncertainty here arises from the measurement process of the instrument, it is classified as a Type A uncertainty. The Type A uncertainty is calculated as follows:

$$unc = \frac{\sigma_x}{\sqrt{n}} = \sqrt{\frac{\sum_{i=1}^{n}(x_i - \bar{x})^2}{n\,(n-1)}},$$ where $\sigma_x$ represents the standard deviation of multiple observations, n denotes the number of observations, $\bar{x}$ is the mean of the observations.

We used multiple measurements obtained during a single observation period, before the tethered balloon was launched, as the observational data to estimate the uncertainty of the black carbon instrument in measuring carbonaceous aerosols. The calculated uncertainty is 0.01793 μg m$^{-3}$.

2. Line 183: Please cite the source of Equation 2.

**Response:** Thank you for your suggestion. We have added the reference (or: citation)

for this formula in the manuscript. The specific revision is as follows:

Additionally, the light absorption coefficient of carbonaceous aerosols can be expressed as:

$$b_{Abs}(\lambda) = k \times \lambda^{-AAE} \tag{2}.$$

In Eq. (2), k is a wavelength-independent constant, and AAE represents the Absorption Ångström Exponent, which characterizes the wavelength dependence of light absorption of carbonaceous aerosols (Ångström, 1929).

Reference

Ångström, A., 1929. On the atmospheric transmission of sun radiation and on dust in the air. Geogr. Ann. 11, 156–166.

3. Line 200: Please cite the source of Equation 4.

**Response:** We appreciate your suggestion. The citation for this formula has now been included in the manuscript. The specific changes are detailed below:

Specific humidity (q, g g$^{-1}$), defined as the ratio of water vapor mass to the total mass of moist air (water vapor plus dry air), is calculated using Eq. (4) (Gutzler, 1992),

$$q = \frac{\varepsilon \times e}{P - 0.378 \times e} \tag{4}$$

Reference

Gutzler, D. S. Climatic variability of temperature and humidity over the tropical western Pacific. Geophys. Res. Lett., 19(15), 1595-1598, https://doi.org/10.1029/92GL01579, 1992.

4. Line 203: Please cite the source of Equation 5.

**Response:** We have adopted your suggestion. The reference for this equation has been incorporated into the main body of the manuscript. The modification is shown below:

Specific humidity (q, g g$^{-1}$), defined as the ratio of water vapor mass to the total mass of moist air (water vapor plus dry air), is calculated using Eq. (4) (Gutzler, 1992),

$$q = \frac{\varepsilon \times e}{P - 0.378 \times e} \tag{4}$$

where $\varepsilon = 0.622$, e represents vapor pressure, and P denotes atmospheric pressure, with

vapor pressure, e being derived from Eq. (5) through relative humidity (RH) and air temperature (T, °C) (Gutzler, 1992).

$$e = 6.105 \times RH \times \exp\left(\frac{17.7 \times T}{237.7 + T}\right) \tag{5}.$$

Reference

Gutzler, D. S. Climatic variability of temperature and humidity over the tropical western Pacific. Geophys. Res. Lett., 19(15), 1595-1598, https://doi.org/10.1029/92GL01579, 1992.

5. Lines 207 – 208: The definition of the term "planetary boundary layer (PBL)" should be provided when the term first appears in the text.

**Response:** Thank you for your suggestion. We have improved the introduction of the PBL in the manuscript to address this issue. The revised content is as follows:

Planetary boundary layer is the part of the atmosphere closest to the planet's surface, accounting for approximately 10% – 20% of the troposphere. It is the lowest layer of the troposphere directly influenced by surface forcing, with a response time of less than one hour, playing a critical role in the dispersion and transport of air pollutants. Within the PBL, turbulent mixing processes homogenize air temperature and humidity, resulting in relatively uniform distributions of these properties.

6. Line 213: Please provide citations to support this statement "various methodologies exist for determining PBL height (PBLH)" or revise the text by simply describing the approaches used in this study. It is recommended to provide citations while stating that various methodologies exist.

**Response:** Thank you for your suggestion. We have revised the text at this point as you recommended. The revised content is as follows:

PBL height (PBLH) refers to the thickness of the layer most significantly affected by the surface. In other words, it is the vertical extent that surface turbulent motion (caused by surface heating, friction, or topography, etc.) can effectively influence (Emeis et al., 2008; Seibert et al., 2000). This study employs two established methodologies for determining the PBLH: the potential temperature gradient method and the parcel

method (Holzworth, 1964; Zhang et al., 2020). The potential temperature gradient method was utilized for calculating the PBLH during the nighttime and early morning hours (20:00, 23:00, 05:00, and 08:00 LT), while the parcel method was applied specifically for the daytime periods (11:00, 14:00, and 17:00 LT). Detailed computational procedures for both approaches are summarized in Supplementary Table S1.

7. Line 227: Please cite the source of Equation 6.

**Response:** We appreciate your suggestion. The citation for this formula has now been included in the manuscript. The specific changes are detailed below:

To quantify the relative contributions of potential temperature gradient, mechanical turbulence index, horizontal wind speed, and vertical wind speed to UVPM variations at different altitudes, we employed a random forest regression algorithm. The model generated training subsets via bootstrap sampling, with random feature subsets selected for optimal splitting at each decision tree node. Observations were categorized into daytime (08:00, 11:00, 14:00, 17:00 LT) and nighttime (20:00, 23:00, 05:00 LT) periods to compare the dominant mechanisms governing aerosol vertical distribution. The mechanical turbulence index was calculated using Eq. (6) (Zhao et al., 2023a).

$$V_{TKE} = 0.5 \times \sqrt{\overline{u^2} + \overline{v^2} + \overline{w^2}} \tag{6}$$

Reference

Zhao, S. P., He, J. J., Dong, L. X., Qi, S. F., Yin, D. Y., Chen, J. B., Yu, Y. Contrasting Vertical Circulation between Severe and Light Air Pollution inside a Deep Basin Results from the Collaborative Experiment of 3D Boundary-Layer Meteorology and Pollution at the Sichuan Basin (BLMP-SCB). Bull. Am. Meteorol. Soc., 104(2), E411-E434, http://doi.org/10.1175/BAMS-D-22-0150.1, 2023a.

8. Lines 245 – 248: It would be great if this qualitative statement can be turned into a table for a quantitative comparison, which would make the argument more convincing and make this work more valuable and academically citable.

**Response:** Thank you for your suggestion. We have added Table 1 in the corresponding section of the manuscript, which makes the comparison in the main text clearer.

Table 1 Statistics of near-surface eBC concentrations in main city.

| City (Research region) | eBC concentrations ($\mu g\ m^{-3}$) | References |
| --- | --- | --- |
| Pingliang | 0.8 (average) | This study |
| Beijing | 2.0 ~ 6.0 | Yang et al., 2022 |
| Shanghai | 6.0 ~ 10.5 | Wang et al., 2021a |
| Nanjing | 3.2 (average) | Shi et al., 2021 |
| Chengdu | 5.0 ~ 8.0 | Zhao et al., 2023a |
| Shenzhen | 1.8 ~ 2.5 | Wu et al., 2021 |
| Hengshui | 5.2 (average) | Ran et al., 2016 |
| Beibu Gulf region | 2.3 ~ 4.0 | Yang et al., 2023 |
| Lanzhou | 3.0 ~ 6.0 | Guan et al., 2022 |

9. Lines 253 – 254: Without showing the quantitative data for direct comparisons of emission inventories of air pollutants from daily human activities, it would be safer to use a more conservative statement like "probably lower" to ensure scientific rigor and avoid controversy.

**Response:** Thank you for your suggestion. I have revised the content in the manuscript accordingly, and the modified content is as follows:

This discrepancy can be attributed to several factors. Firstly, Pingliang is a relatively small city with a permanent population of fewer than 2 million, whereas the aforementioned cities have much larger urban populations. Consequently, the total amount of air pollutants generated from daily human activities in Pingliang may be comparatively lower than those in the aforementioned cities.

10. Line 288: It seems that only diurnal variations of IRBC rather than BC are shown.

**Response:** Thank you for your suggestion. We have already addressed similar issues consistently throughout the manuscript. We have defined the concentration measured at the 880 nm wavelength (IR) as equivalent Black Carbon (eBC) and have made corresponding revisions throughout the entire text.

3.1.2 Diurnal variations of eBC and UVPM profiles

Figures S3 and S4 respectively depict the trends and correlations of the ascent and descent profiles for eBC and UVPM. The results indicate that the ascent and descent profiles at 5:00, 20:00, and 23:00 in this study exhibit similar trends, and thus they can both be treated as independent observational profiles for analysis. For observations at 8:00, 11:00, 14:00 and 17:00, only the ascent data were used for analysis. Figure 2 shows the averaged profiles for all sampling periods during the observation campaign, the red solid line and its shaded envelope denote the mean and standard deviation of eBC, while the blue solid line and its shaded envelope denote the mean and standard deviation of UVPM. Because eBC mass concentration in the upper atmosphere is extremely low (below the instrument's normal limit of detection), optical measurements often yield negative results, which are corrected when the ONA method is used for data quality control. Furthermore, owing to the relatively high wind speeds in the upper atmosphere, our measurements in the early afternoon were typically unable to reach the PBLH.

11. Line 290: Are the profiles for BC or IRBC?

**Response:** Thank you for your suggestion. We have clarified this issue consistently throughout the revised manuscript. The concentration measured at 880 nm has been defined as equivalent black carbon (eBC), and all relevant terms have been updated accordingly across the entire text.

12. Line 295: The term IRBC is used here while referring to BC mentioned earlier (Line 290). It seems that IRBC only represents black carbon (BC) measured at wavelength 880 nm while the absorption measured at wavelengths 470nm, 528nm, 625 nm, and 880 nm are all considered as black carbon in this study. If this is the case, please revise the definition in earlier sections of this manuscript for consistency.

**Response:** Thank you for your suggestion. We have addressed similar issues consistently throughout the manuscript. The concentrations measured at wavelengths of 375 nm, 470 nm, 528 nm, and 625 nm correspond to carbonaceous aerosols,

including black carbon (BC), brown carbon (BrC). In this study, to distinguish carbonaceous aerosols measured at different wavelengths, the concentration measured at 375 nm is referred to as ultraviolet-absorbing particulate matter (UVPM), while the concentration measured at 880 nm is defined as equivalent black carbon (eBC). Corresponding revisions have been made throughout the manuscript. The specific changes are as follows:

In Figure 2, the red solid line and its shaded envelope denote the mean and standard deviation of eBC, while the blue solid line and its shaded envelope denote the mean and standard deviation of UVPM.

[Figure]

Figure 2. Averaged diurnal variation in profiles of the eBC and UVPM concentration during the field campaign. The red and blue lines represent eBC and UVPM concentrations, respectively, while the red and blue shaded areas denote the standard deviation of eBC and UVPM, respectively. The light orange shaded area represents the difference between UVPM and eBC.

13. Line 300: Are the profiles for BC or IRBC?

**Response:** Thank you for your suggestion. We have already addressed similar issues consistently throughout the manuscript. We have defined the concentration measured at the 880 nm wavelength as equivalent Black Carbon (eBC) and have made corresponding revisions throughout the entire text. The specific modification at this point is as follows:

Vertical profiles of eBC and UVPM concentrations reveal a consistent decrease with increasing altitude, with the most pronounced gradient observed in the near-surface layer.

[Figure]

Figure 2. Averaged diurnal variation in profiles of the eBC and UVPM concentration during the field campaign. The red and blue lines represent eBC and UVPM concentrations, respectively, while the red and blue shaded areas denote the standard deviation of eBC and UVPM, respectively. The light orange shaded area represents the difference between UVPM and eBC.

14. Line 304: Is the BC here meant to be IRBC?

**Response:** Thank you for your suggestion. We have already addressed similar issues consistently throughout the manuscript. We have defined the concentration measured at the 880 nm wavelength as equivalent Black Carbon (eBC) and have made corresponding revisions throughout the entire text. The specific modification at this point is as follows:

Vertical profiles of eBC and UVPM concentrations reveal a consistent decrease with increasing altitude, with the most pronounced gradient observed in the near-surface layer. A comparative analysis of their vertical profiles reveals that during periods of weak convective activity, such as early morning and nighttime (i.e., 05:00, 20:00, and 23:00 LT), UVPM concentrations aloft generally exceed those of eBC (the significance test of the UVPM–eBC differences is provided in Figure S5), while near the surface the

two species show much smaller differences.

[Figure]

Figure S5. Significance test of the UVPM–eBC differences during different hours of the day. The red and blue dashed lines in the figure indicate p-values of 0.05 and 0.01, respectively.

15. Line 305: Is the UVPM-to-BC here meant to be UVPM-to-IRBC?

**Response:** Thank you for your suggestion. The term IR BC (Infrared Black Carbon) used here refers to the concentration measured at the 880 nm wavelength, which is defined as equivalent Black Carbon (eBC). We have revised all related terminology throughout the manuscript. The specific modification at this point is as follows:

Vertical profiles of eBC and UVPM concentrations reveal a consistent decrease with increasing altitude, with the most pronounced gradient observed in the near-surface layer. A comparative analysis of their vertical profiles reveals that during periods of weak convective activity, such as early morning and nighttime (i.e., 05:00, 20:00, and 23:00 LT), UVPM concentrations aloft generally exceed those of eBC (the significance test of the UVPM–eBC differences is provided in Figure S5), while near the surface the two species show much smaller differences.

[Figure]

Figure S5. Significance test of the UVPM–eBC differences during different hours of the day. The red and blue dashed lines in the figure indicate p-values of 0.05 and 0.01, respectively.

16. Line 316 – 317: Could this change due to vertical mixing and dominance of black carbon? The disparity between UVPM and IRBC should not disappear at lower altitudes if the pollutants considered as UVPM mixed downward during the convection do not include black carbon. If UVPM is considered as the total absorption of dust, brown carbon, and black carbon, this reduction of disparity might indicate that the contribution of dust and organics (brown carbon) becomes less important, and IRBC is dominant during this period.

**Response:** Thank you for your question and suggestion. We have revised this section and added supporting evidence. According to the mean diurnal variations in vertical distribution of the $UVPM_{sec}/UVPM$ ratio, the larger nighttime difference between UVPM and eBC at higher altitudes is primarily attributable to the increased contribution of $UVPM_{sec}$. The relevant discussion is presented below:

Vertical profiles of eBC and UVPM concentrations reveal a consistent decrease with increasing altitude, with the most pronounced gradient observed in the near-surface layer. A comparative analysis of their vertical profiles reveals that during periods of weak convective activity, such as early morning and nighttime (i.e., 05:00, 20:00, and 23:00 LT), UVPM concentrations aloft generally exceed those of eBC (the significance test of the UVPM–eBC differences is provided in Figure S5), while near the surface the

two species show much smaller differences. To further investigate the underlying causes, we estimated the concentration of secondary UVPM ($UVPM_{sec}$) using the least-squares method described by Wu et al. (2016), based on the ratio of UVPM to eBC. Figure 3 presents the mean diurnal variations in vertical distributions of the ratios of $UVPM_{sec}$ to total UVPM, and of $UVPM_{sec}$ to eBC. From 17:00 to 05:00 LT, the $UVPM_{sec}$/UVPM and $UVPM_{sec}$/eBC ratios aloft increase markedly, indicating that the elevated UVPM during this period is more strongly influenced by secondary sources. As time progresses, the region characterized by high ratios gradually approaches the ground surface and eventually disappears by 08:00 LT.

The enhanced contribution of secondary sources aloft may be attributed to the increasingly stable stratification after 17:00 LT, which suppresses vertical mixing and inhibits the upward transport of both eBC and UVPM. In contrast, gaseous precursors of $UVPM_{sec}$ (i.e., VOCs) can accumulate above the PBL. Under relatively clean atmospheric conditions with limited condensation sinks, nocturnal chemistry dominated by $NO_3^-$ radicals, together with low ambient temperatures, favors the formation of secondary organic aerosols through gas-to-particle partitioning and temperature-dependent condensation (Han & Jang, 2023; Kuang et al., 2025; Kulmala, 2022; Morgan et al., 2009; Wang et al., 2023; Zhao et al., 2023b). After sunrise (around 06:00 LT in summer at this site), enhanced solar radiation promotes convective mixing and weakens the nocturnal inversion at the top of the PBL. Consequently, $UVPM_{sec}$-enriched air masses retained within the residual layer are entrained downward into the growing daytime PBL, thereby reducing the concentration gradient between UVPM and eBC within the PBL (Zhao et al., 2023a). In addition, increased human activities and strengthened thermal convection after sunrise lead to larger primary emissions throughout the atmospheric column, causing carbonaceous aerosols to be increasingly dominated by primary components. Because the PBL is still developing at 08:00 LT, the near-surface concentrations of UVPM and eBC are higher than those observed at 05:00 LT. As thermal convection intensifies, the decline in near-surface UVPM and eBC slows between 11:00 and 14:00 LT. Between 14:00 and 17:00 LT, strong

convective mixing results in an approximately uniform vertical distribution of both eBC and UVPM from the surface up to ~800 m.

[Figure]

Figure 3. Mean diurnal variations in vertical distribution of a) UVPM$_{sec}$/UVPM and b) UVPM$_{sec}$/eBC ratios during the field campaign.

References

Han, S., Jang, M. Modeling daytime and nighttime secondary organic aerosol formation via multiphase reactions of biogenic hydrocarbons. Atmos. Chem. Phys., 23(2), 1209-1226, http://doi.org/10.5194/acp-23-1209-2023, 2023.

Kuang, Y., Luo, B., Huang, S., Liu, J. W., Hu, W. W., Peng, Y. W., Chen, D. H., Yue, D. L., Xu, W. Y., Yuan, B., Shao, M. Formation of highly absorptive secondary brown carbon through nighttime multiphase chemistry of biomass burning emissions. Atmos. Chem. Phys., 25(6), 3737-3752, http://doi.org/10.5194/acp-25-3737-2025, 2025.

Kulmala, M., Cai, R., Stolzenburg, D., Zhou, Y., Dada, L., Guo, Y., Yan, C., Petäjä, T., Jiang, J., Kerminen, V. M: The contribution of new particle formation and subsequent growth to haze formation, Environ. Sci.: Atmos., 2: 352–361, https://doi.org/10.1039/D1EA00096A, 2022.

Morgan, W. T., Allan, J. D., Bower, K. N., Capes, G., Crosier, J., Williams, P. I., Coe, H. Vertical distribution of sub-micron aerosol chemical composition from North-Western Europe and the North-East Atlantic. Atmos. Chem. Phys., 9, 5389–5401, https://doi.org/10.5194/acp-9-5389-2009, 2009.

Wang, H. C., Wang, H. L., Lu, X., Lu, K. D., Zhang, L., Tham, Y. J., Shi, Z. B., Aikin, K., Fan,

S. J., Brown, S. S., Zhang, Y. H. Increased night-time oxidation over China despite widespread decrease across the globe. Nat. Geosci., 16: 217-223, https://doi.org/10.1038/s41561-022-01122-x, 2023.

Zhao, S. P., Yu, Y., He, J. J., Dong, L. X., Qi, S. F. A New Physical Mechanism of Rainfall Facilitation to New Particle Formation. Geophys. Res. Lett., 51(1), e2023GL106842, https://doi.org/10.1029/2023GL106842, 2023.

Wu, C. and Yu, J. Z.: Determination of primary combustion source organic carbon-to-elemental carbon (OC / EC) ratio using ambient OC and EC measurements: secondary OC-EC correlation minimization method. Atmos. Chem. Phys., 16, 5453-5465, doi:10.5194/acp-16-5453-2016, 2016.

17. Lines 321 – 324: This observation also indicates that the UVPM defined in this study is probably the overall contribution of black carbon, brown carbon, and dust aerosol. Please clarify the definition.

**Response:** Thank you for raising this question. The definition of UVPM has been revised in Section 2.1.2 of the updated manuscript. According to Section 1.2 of the official manual for the MicroAeth® MA series MA350 instrument, substances measured at a wavelength of 375 nm are defined as UVPM. Since the observation period corresponds to the local summer season, except for a specific dust event, the regional air quality was generally good without significant dust pollution. Based on the instrument manual (URL: https://aethlabs.com/products/ma300) and previous studies (Zhao et al., 2023a), UVPM in this study comprises black carbon (BC) and brown carbon. The relevant discussion is presented below:

The instrument employs five laser wavelengths: 375 nm (UV), 470 nm (Blue), 528 nm (Green), 625 nm (Red), and 880 nm (IR). Carbon concentrations measured at 880 nm are considered to represent the equivalent black carbon concentration (denoted as eBC) (Zhao et al., 2023a), whereas those measured at 375 nm correspond to ultraviolet-absorbing particulate matter (UVPM), which includes primary brown carbon and black carbon emitted from sources such as biomass burning and coal combustion, as well as

secondary brown carbon formed through atmospheric oxidation processes (The relevant introduction/details regarding the substance measured at a wavelength of 375 nm can be referenced in the official manual of the MicroAeth® MA Series MA300 instrument. URL: https://aethlabs.com/products/ma300, last accessed: 13 November 2025).

Reference

Zhao, S. P., He, J. J., Dong, L. X., Qi, S. F., Yin, D. Y., Chen, J. B., Yu, Y. Contrasting Vertical Circulation between Severe and Light Air Pollution inside a Deep Basin Results from the Collaborative Experiment of 3D Boundary-Layer Meteorology and Pollution at the Sichuan Basin (BLMP-SCB). Bull. Am. Meteorol. Soc., 104(2), E411-E434, http://doi.org/10.1175/BAMS-D-22-0150.1, 2023a.

18. The term "LT" is used in Lines 225 – 226 while the term "LST" is used in other parts of the manuscript. Please explicitly distinguish the difference between the two terms. Otherwise, please choose one and stick to it to ensure consistency and avoid confusion.

**Response:** Thank you for pointing out this inconsistency. We agree that using different abbreviations for the same concept may cause confusion. The term "LT" used in lines 225–226 refers to "Local time"; however, its usage was inconsistent with other parts of the manuscript. To ensure uniformity throughout the paper, we have revised all instances of "LST" to "LT".

19. Figure 3: It seems that IRBC rather than BC is shown in the figure. Please update the caption and revise relevant statements in the main text.

**Response:** Thank you for your suggestion. We have already addressed similar issues consistently throughout the manuscript. We have defined the concentration measured at the 880 nm wavelength as equivalent Black Carbon (eBC) and have made corresponding revisions throughout the entire text. The specific modification at this point is as follows (Due to adjustments made to the figures within the manuscript, this illustration has been renumbered as "Figure 4."):

[Figure]

Figure 4. Cluster analysis of UVPM and eBC concentration profiles during the field campaign. The blue, red, and green lines represent the profiles of UVPM, eBC, and potential temperature, respectively, and the shaded areas in the corresponding colors denote their standard deviations.

20. Section 3.1.3: It seems that the BC used in this section should be IRBC. Please confirm.

**Response:** Thank you for your suggestion. We have already addressed similar issues consistently throughout the manuscript. We have defined the concentration measured at the 880 nm wavelength as equivalent Black Carbon (eBC) and have made corresponding revisions throughout the entire text.

21. Lines 364 – 366: How are the weather conditions defined based on what parameters and what data? Please provide relevant information.

**Response:** Thank you for your comment. We have added detailed information about the environmental conditions in the Supplementary Material. The specific content is provided below:

We further selected observations from representative pollution events to compare AAE under different environmental conditions (Figure 5) (Detailed explanations regarding the selection of pollution events are provided in Text 3 of the supplementary material).

**Text 3: Selection guidelines for pollution incidents**

1. Diesel vehicle emissions: During the 20:00 observation on July 17, 2024, a diesel harvester was operating approximately 50 m from our observation site and ceased work after about half an hour. Consequently, we define this specific observation as being affected by diesel vehicle emissions. For comparison, we selected the 23:00

observation—which has the shortest temporal difference—as the corresponding non-diesel vehicle emission observation.

2. Dust Pollution: A dust event occurred throughout the entire day of July 21, 2024. We defined the 8:00 observation, when the dust was most severe ($PM_{10}$ concentration in Pingliang urban area was 350 $\mu g\ m^{-3}$ at that time), as the dust pollution observation. Since the dust event lasted for a long duration, we chose the observation taken at the same time on the previous day (8:00 on July 20, 2024) for comparison.

The relevant information mentioned above is now reflected using shading and text annotations in the new Figure S1.

[Figure]

Figure S1. Temporal variations of meteorological parameters, including air temperature (T), relative humidity (RH), wind direction (WD), wind speed (WS), and air pollutants, including particulate matter with aerodynamic diameter smaller than 2.5 $\mu m$ ($PM_{2.5}$) and than 10 $\mu m$ ($PM_{10}$), nitrogen dioxide ($NO_2$), ozone ($O_3$), sulfur dioxide ($SO_2$), and carbon monoxide (CO), were observed during the monitoring period. The shaded areas in the figure represent the during and after vehicle emissions, no dust pollution, and dust pollution at 20:00 on July 17, 23:00 on July 17, 2024, 08:00 on July 20, 2024, 08:00 on July 21, 2024, respectively.

22. Lines 375 – 377: Please provide citations of previous studies.

**Response:** Thank you for pointing out this issue. We have added citations to previous studies in the manuscript. The revised content is as follows:

Previous studies commonly employ a two-component model to differentiate the Absorption Ångström Exponent of eBC ($AAE_{eBC}$) and BrC ($AAE_{UVPM}$), fixing $AAE_{eBC}$ at 1 and thereby deriving $AAE_{UVPM}$ (Figure S9, the calculation methodology is provided in Text 4 of the supplementary material) (Chen et al., 2015; Chow et al., 2018). Elevated values of $AAE_{UVPM}$ are typically indicative of biomass burning and secondary aging processes (Olson et al., 2015; Gombi et al., 2025).

References

Chen, L.W.A., Chow, J.C., Wang, X.L., Robles, J.A., Sumlin, B.J., Lowenthal, D.H., Zimmermann, R., Watson, J.G. Multi-wavelength optical measurement to enhance thermal/optical analysis for carbonaceous aerosol. Atmos. Meas. Tech., 8(1), 451-461, https://doi.org/10.5194/amt-8-451-2015, 2015.

Chow, J.C., Watson, J.G., Green, M.C., Wang, X.L., Chen, L.-W.A., Trimble, D.L., Cropper, P.M., Kohl, S.D., Gronstal, S.B. Separation of brown carbon from black carbon for IMPROVE and CSN PM2.5 samples, Air Waste Manag. Assoc., 68(5), 494-510, https://doi.org/10.1080/10962247.2018.1426653, 2018.

Gombi, C., Rahman, A., Hodovány, S. et al. A demonstrative study of a novel approach for spectral based source apportionment of ambient aerosols. Sci. Rep., 15, 19501, https://doi.org/10.1038/s41598-025-04022-3, 2025.

Olson, M. R., Victiria, G. M., Robinson, M. A., Rooy, P. V., Dietenberger, M. A., Bergin, M., Schauer, J. J. Investigation of black and brown carbon multiple-wavelength-dependent light absorption from biomass and fossil fuel combustion source emissions. J. Geophys. Res.-Atmos., 120(13): 6682-6697, https://doi.org/10.1002/2014JD022970, 2015.

23. Lines 407 – 408 and Lines 414 – 416: The potential-temperature gradient method is stated to yield accurate estimates at night and in the early morning in Lines 407 – 408. However, the potential-temperature gradient method is stated to overshoot the inversion top in Lines 414 – 416, which seems to be contradictory with the earlier statement of accurate estimates. Please revise the statements to clarify and avoid confusion.

**Response:** Thank you for pointing out this issue. As our study does not involve the

PBLH$_C$, we have removed the related statements from the manuscript.

24. Figure 5: Why are the PBLH (Parcel) lines only shown in some of the panels in the figure while the PBLH (Gradient) lines are shown in all panels? Please clarify in the revision.

**Response:** Due to adjustments made to the figures within the manuscript, this illustration has been renumbered as Figure 6. Based on Holzworth (1964)'s definition of the Parcel Method as: "The height at which a dry adiabatic line extending upward from the maximum surface temperature intersects the most recently observed temperature profile," it is more suitable for calculating the PBLH (Planetary boundary layer height) in unstable atmospheric conditions during the daytime. However, this method cannot calculate PBLH under stable nocturnal stratification conditions. Consequently, the PBLH calculated by the Parcel Method is not displayed in the figures representing the nighttime periods. Furthermore, during the 14:00 observation period, the observation height did not reach the actual PBLH, and therefore, the PBLH calculated by the Parcel Method is also not shown. Although the potential temperature gradient method can be used to calculate PBLH at every observation time, its results are more reliable under stable stratification, which is why it was used in this study to identify the PBLH for the (20:00, 23:00, 05:00, and 08:00) time slots. This explanation can be found in the Section 2.3 and first paragraph of Section 3.2 of the manuscript. In addition, we revised Figure 6 by retaining only the PBLH derived from the observational profiles. The specific content is as follows:

In Section 2.3:

Planetary boundary layer is the part of the atmosphere closest to the planet's surface, accounting for approximately 10% – 20% of the troposphere. It is the lowest layer of the troposphere directly influenced by surface forcing, with a response time of less than one hour, playing a critical role in the dispersion and transport of air pollutants. Within the PBL, turbulent mixing processes homogenize air temperature and humidity, resulting in relatively uniform distributions of these properties. PBL height (PBLH)

refers to the thickness of the layer most significantly affected by the surface. In other words, it is the vertical extent that surface turbulent motion (caused by surface heating, friction, or topography, etc.) can effectively influence (Emeis et al., 2008; Seibert et al., 2000). This study employs two established methodologies for determining the PBLH: the potential temperature gradient method and the parcel method (Holzworth, 1964; Zhang et al., 2020). The potential temperature gradient method was utilized for calculating the PBLH during the nighttime and early morning hours (20:00, 23:00, 05:00, and 08:00 LT), while the parcel method was applied specifically for the daytime periods (11:00, 14:00, and 17:00 LT). Detailed computational procedures for both approaches are summarized in Supplementary Table S1.

In Section 3.2:

The parcel method is well suited to convective conditions but cannot accurately resolve PBLH during early morning and nighttime (Holzworth, 1964), whereas it performs reliably under strong daytime convection. Using this approach, PBLH at 11:00 and 17:00 LT were 508 m and 450 m, respectively; at 14:00, measurement ceilings did not reach PBLH. By contrast, the potential temperature gradient method yields accurate estimates at night and in the early morning: PBLH at 05:00, 08:00, 20:00, and 23:00 LT were 260 m, 181 m, 260 m, and 223 m, respectively.

[Figure]

Figure 6. Diurnal variations in profiles of eBC, UVPM and potential temperature on July 27, 2024. The red line represents eBC, the blue line represents UVPM, and the green line represents the potential temperature profile. The light orange shaded area represents the difference between UVPM and eBC. The black dashed lines represent the PBLH.

References

Holzworth, G. C. Estimates of mean maximum mixing depths in the contiguous united states. Mon. Weather Rev., 92(5), 235-242, http://doi.org/10.1175/ 1520-0493(1964)092<0235:EOMMMD>2.3.CO;2, 1964.

Zhang, H. S., Zhang, X. Y., Li, Q. H., Cai, X. H., Fan, S. J., Song, Y., Hu, F., Che, H. Z., Quan, J. N., Kang, L., Zhu, T. Research progress on estimation of atmospheric boundary layer height. Acta Meteorol. Sin., 78(3), 522-536, http://doi.org/10.11676/qxxb2020.044, 2020.

25. Lines 504 – 505: Point sources have continuous emissions. How can they be linked to high concentrations only infrequently or attributed to the cause of infrequently occurred high concentrations under the same wind direction? Please provide supporting information or data to strengthen this statement.

**Response:** Thank you for your valuable comment. We have carefully reviewed the manuscript content. Given that our observation site is located in a rural area with no stationary point sources. Sources of pollution nearby, the occasional high pollution concentrations observed under the same wind direction are likely associated with the non-scheduled production activities of local residents. Consequently, pollutants are not continuously transported to the observation site under a constant wind direction. We have revised the term "local point sources" to the more appropriate term "local and sporadic emissions" throughout the manuscript. The specific modification regarding this observation is as follows:

Southeasterly winds occasionally lead to relatively high UVPM values at both 100 m and 500 m altitude, although this occurrence is less frequent, which is consistent with the results of backward trajectories. We hypothesize that this variability may be associated with local and sporadic emissions from residents in the vicinity, as the observation site is located in a rural area with no stationary point sources. However, the specific cause cannot be definitively determined by the CPF alone. We will continue to

analyze this phenomenon in subsequent observational studies to provide a clearer explanation.

26. Lines 567 – 568: What about 14:00?

**Response:** Thank you for your suggestion. The reason that the 14:00 observation period was not included in this description is that the number of soundings collected at 14:00 during the campaign was insufficient (or limited), and none of them reached the corresponding PBLH for that time slot. This has already been mentioned in section 3.2 of the manuscript. The original text is as follows:

At 14:00 LT tethered-balloon sampling did not reach the PBL top, but observations within the PBL show uniform aerosol distributions, preventing a direct assessment of PBLH effects on concentration profiles.

27. Lines 570 – 572: Please double check the logic of this statement. It seems to be meant to contrast between thermodynamic and dynamic processes while the thermodynamic process is mentioned twice here.

**Response:** Thank you for pointing out this issue. Here, we made a writing error; the original intent should have been: "Consequently, the vertical distribution of carbonaceous aerosols within the daytime PBL is primarily governed by thermodynamic processes, in contrast to the combined dynamic and thermodynamic control that dominates within the nocturnal residual layer." We have made the correction in the manuscript, and the revised content is as follows:

Consequently, the vertical distribution of carbonaceous aerosols within the daytime PBL is primarily governed by thermodynamic processes, in contrast to the combined dynamic and thermodynamic control that dominates within the nocturnal residual layer.

28. Figure S2: Is there a specific definition of the term "conventional air pollutants"? It would be great if the caption could be specifically listing the air pollutants shown in this figure.

**Response:** Thank you for your comment. We have merged Figures S1 and S2 into a

[Figure]

Figure S1. Temporal variations of meteorological parameters, including air temperature (T), relative humidity (RH), wind direction (WD), wind speed (WS), and air pollutants, including particulate matter with aerodynamic diameter smaller than 2.5 μm ($PM_{2.5}$) and than 10 μm ($PM_{10}$), nitrogen dioxide ($NO_2$), ozone ($O_3$), sulfur dioxide ($SO_2$), and carbon monoxide (CO), were observed during the monitoring period. The shaded areas in the figure represent the during and after vehicle emissions, no dust pollution, and dust pollution at 20:00 on July 17, 23:00 on July 17, 2024, 08:00 on July 20, 2024, 08:00 on July 21, 2024, respectively.

29. Figures S3 and S4: The terms "ascent" and "descent" are used in the main text and figure caption while the terms "rising" and "landing" are used in the figure legend, which are not consistent with each other. It would be great if the terminology could be consistent throughout the work.

**Response:** Thank you for your suggestion. We have revised this issue, and the modified content is as follows:

[Figure]

Figure S3. Comparison of equivalent black carbon (eBC) vertical profiles during ascent with those during descent of the tethered balloon.

[Figure]

Figure S4. Comparison of ultraviolet-absorbing particulate matter (UVPM) vertical profiles during ascent with those during descent of the tethered balloon.

**Technical Comments:**

1. Line 23: The parameter "VTKE" seems to be a typographical error of "$V_{TKE}$".

**Response:** Thank you for pointing out this shortcoming. We have made the corresponding revisions in the manuscript. The modified content is as follows:

A comparison of the vertical profiles of eBC, UVPM, $V_{TKE}$ (mechanical turbulence), and potential temperature showed that during the early morning and nighttime, when convective activity was weak, UVPM concentrations in the upper atmosphere were higher than those of eBC.

2. Line 413: The word "and" in front of the word "often" seems to be a typographical error and should be deleted.

**Response:** Thank you for pointing out this shortcoming. Since this study does not involve PBLHc, this part has been removed from the manuscript.

3. Line 637: The word "understanded" seems to be a typographical error of "understood".

**Response:** Thank you for pointing out this error. We have made the correction in the manuscript. The revised content is as follows:

Furthermore, the filed campaign was conducted at only a site, and thus the feedback between aerosols and the PBL meteorology cannot be fully understood at whole Loess Plateau.